# Divide and Conquer: Selective Value Learning and Policy Optimization for Offline Safe Reinforcement Learning

**Jiahui Zhu** *School of EECS, Washington State University*              *jiahui.zhu@wsu.edu*

**Lei Ying** *Department of EECS, University of Michigan*              *leiying@umich.edu*

**Honghao Wei** *School of EECS, Washington State University*              *honghao.wei@wsu.edu*

**Reviewed on OpenReview:** *https: // openreview. net/ forum? id=4KYrv6qYMl*

## Abstract

Offline safe reinforcement learning (RL) aims to learn policies that maximize reward while satisfying safety constraints from a fixed dataset. In practice, safety-critical offline datasets are often *mixed-quality*, containing both safe/high-quality trajectories and unsafe/low-quality redtrajectories. Existing methods extend offline RL with primal–dual value learning and behavior-regularized policy optimization, but under mixed-quality data they struggle: uniform updates across all states ignore the difference between safety-preserving and unsafe states, leading to inaccurate value estimates, infeasible solutions when constraints conflict, and strong sensitivity to dataset quality. We propose *SEVPO* (SElective Value Learning and Policy Optimization), a divide-and-conquer framework that separates updates based on state safety. SEVPO learns conservative cost values to identify safe states, applying reward-constrained optimization with selective regularization within them, and switching to cost minimization elsewhere to compute least-cost escape paths. Extensive experiments across diverse mixed-quality datasets show SEVPO achieves high reward while providing strict safety guarantees, outperforming state-of-the-art offline safe RL methods across diverse dataset qualities. We further validate SEVPO by training a simulated Unitree Go2 quadruped robot in dynamic environments using only offline data, demonstrating its potential for safety-critical robotics (`https://youtu.be/tDpWq2EV_Ig`). The source code is publicly available at: `https://github.com/JiahuiZhu666/SEVPO/tree/main`

## 1 Introduction

Safe reinforcement learning (RL) has been successfully applied in areas such as autonomous driving (Isele et al., 2018), recommender systems (Chow et al., 2017), and robotics (Achiam et al., 2017; Dawson et al., 2023), helping to develop policies that adhere to critical safety constraints, including collision avoidance, budget management, and reliability requirements. However, a significant challenge lies in gathering the data needed to train these policies in an online setting. In many real-world applications, interacting with the environment is not only costly and time-consuming but also inherently risky. Training an RL agent online requires learning through trial and error, which can lead to unsafe behavior during the learning process, an unacceptable risk in safety-critical tasks.

To overcome the limitations of data collection, offline RL offers a safer alternative by enabling policies to be learned from pre-collected data, without the need for direct interaction with the environment during training. In safety-critical domains, however, these datasets are often *mixed-quality*: they mix safe, near-feasible trajectories with unsafe or suboptimal behaviors collected under different controllers, operating conditions, or exploration policies. This setting is particularly challenging because the agent must improve performance and satisfy strict safety constraints *solely* from a fixed dataset, with no opportunity for additional safe exploration to correct data deficiencies. A central technical obstacle in offline RL is distribution shift: when

the learned policy selects actions that are out-of-distribution (OOD) relative to the dataset (Fujimoto et al., 2019), value estimates can suffer from extrapolation error, leading to unreliable policy improvement and potentially unsafe behaviors.

Mitigating the impact of OOD actions is a critical challenge in the design of offline RL. A common approach is to penalize $Q$-values for OOD actions (Kumar et al., 2020; Yu et al., 2021), producing conservative value estimates that help suppress erroneously high predictions and impose an implicit policy constraint. While effective at reducing overestimation, these methods often suffer from overly conservative value functions, which can severely limit policy improvement. An alternative line of work (Chemingui et al., 2025; Fujimoto et al., 2019; Fujimoto and Gu, 2021; Peng et al., 2019; Siegel et al., 2020; Wang et al., 2020; Jaques et al., 2019; Wu et al., 2019; Dadashi et al., 2021; Zhang and Tan, 2024) imposes explicit policy constraints during evaluation or improvement to avoid unsafe OOD actions. However, safe offline RL introduces challenges that go well beyond those in standard offline RL: The agent must simultaneously maximize reward and satisfy strict safety requirements using only a fixed dataset. In safe offline RL, this dataset often provides limited coverage of safe transitions while also containing unsafe or suboptimal behavior. As a result, even methods that impose explicit policy constraints can be insufficient: without adequate safe coverage, the dataset may not support identifying actions that both improve reward and remain safe, and since no additional exploration is possible, the agent cannot correct or compensate for these dataset deficiencies.

Most existing methods extend standard offline RL with primal–dual formulations (Tessler et al., 2018; Chow et al., 2017; Ding et al., 2020; Efroni et al., 2020; Wei et al., 2022a;a; Kumar et al., 2019; Zhu et al., 2025; Chemingui et al., 2025), converting the constrained problem into an unconstrained one by adding the cost as a penalty to the reward. While conceptually simple, these approaches are highly sensitive to the choice of the Lagrangian multiplier, making it difficult to strike the right balance between reward and safety, especially when the dataset distribution varies. Other offline safe RL algorithms take different routes: CPQ (Xu et al., 2022) leverages a VAE to detect and penalize OOD actions based on their costs; COptiDICE (Lee et al., 2022), an extension of OptiDICE (Lee et al., 2021), optimizes the policy in the stationary distribution space under a cost constraint; decision-transformer approaches (Chen et al., 2021; Liu et al., 2023b) and diffusion-based methods (Janner et al., 2022) update policies at the trajectory level; and VOCE (Guan et al., 2024) addresses OOD extrapolation through pessimistic $Q$-value estimation for both rewards and costs. Despite their differences, these methods share a common limitation: they perform value learning and policy optimization *uniformly* across the entire state space. This uniform treatment implicitly assumes that the same update rule can guarantee safety on average, but it fails to account for the fundamentally different roles of safe and unsafe states. In practice, this means that policies may satisfy constraints in expectation while still violating safety at specific states, which is unacceptable in safety-critical applications. FISOR (Zheng et al., 2024) takes a step forward by decoupling reward maximization and safety constraint satisfaction under different feasibility conditions. However, it still applies a single optimization objective across all states. As we will show later, this global update can lead to infeasible solutions when the dataset contains unsafe or conflicting transitions, underscoring the need for a selective, state-aware approach to value learning and policy optimization in offline safe RL.

These challenges motivate our divide-and-conquer approach, *SEVPO* (SElective Value Learning and Policy Optimization), which separates value learning and policy optimization for safe and unsafe regions. SEVPO estimates the cost value function under the most conservative policy to identify the largest safe set, applying reward-constrained optimization inside it and cost-minimization outside to find the least-cost escape path. Unlike existing offline safe RL methods with uniform objectives(FISOR (Zheng et al., 2024)), which apply behavior regularization uniformly and can inadvertently propagate unsafe behaviors under mixed-quality data, SEVPO selectively applies policy regularization only where it stabilizes learning, avoiding infeasible solutions and enabling efficient recovery. Figure 1 illustrates this on a toy navigation task with varying dataset quality. SEVPO maintains safety and high reward across all scenarios, while the state-of-the-art FISOR (Zheng et al., 2024) fails as the safe-to-unsafe ratio decreases. Our contributions are summarized as follows:

- SEVPO addresses the infeasibility of uniform offline constraints through a principled, region-aware framework that jointly performs selective value learning and policy optimization. Using offline trajectories, it partitions the state space into learned safe and unsafe regions and applies tailored

objectives in each. In the safe region, SEVPO learns reward and cost value functions under the most conservative policy and maintains a KL proximity to the behavior policy to enable stable reward maximization under multiple constraints. In unsafe states, it switches both value learning and policy updates to focus purely on minimizing cumulative cost and removes behavior regularization entirely, avoiding bias from unsafe or suboptimal data. This adaptive divide-and-conquer treatment keeps the optimization problem feasible where uniform baselines often fail, yielding higher rewards in safe regions and lower costs in unsafe ones.

- Extensive experiments on the DSRL benchmark (Liu et al., 2023a) show that SEVPO significantly outperforms state-of-the-art baselines in reward maximization and constraint satisfaction. On Safety-Gymnasium tasks (Ray et al., 2019; Ji et al., 2023), SEVPO increases rewards by **5%** and reduces costs by **52%** on average compared to the best baseline (Table 1). Additionally, Table 1 shows that SEVPO consistently maintains safety while others fail as the data quality degrades. Ablation studies evaluate its performance under different cost limits. We further demonstrate the practical applicability of SEVPO in a safety-critical navigation task by training a Unitree Go2 quadruped robot in the Isaac Gym(Makoviychuk et al., 2021) simulator using only pre-collected offline data. SEVPO successfully learns high-performing policies capable of handling complex, dynamic environments, highlighting its potential for real-world robotic deployment where exploration is costly or unsafe.

- To explore how to **safely** learn a safe policy, we propose a train-collect-and-train framework, which significantly enhances SEVPO's performance when new data collection is allowed. With just one round of new data collection, performance improved by **72%** with a policy safety guarantee.

## 2 Preliminary

We formulate safe RL as a Constrained Markov Decision Process (CMDP), denoted by $\mathcal{M} = (\mathcal{S}, \mathcal{A}, \gamma, \mathcal{P}, r, c, \rho)$. Here, $\mathcal{S}$ and $\mathcal{A}$ are the state and action spaces, $\rho$ is the initial state distribution, and $\gamma \in [0, 1)$ is the discount factor. The reward function is $r : \mathcal{S} \times \mathcal{A} \to \mathbb{R}$, and the cost function is $c : \mathcal{S} \times \mathcal{A} \to \mathbb{R}_+$. We use $c = 0$ to indicate that the state constraint is satisfied. At each time step, the agent observes $s \in \mathcal{S}$, takes $a \in \mathcal{A}$ according to policy $\pi$, receives reward $r(s, a)$ and cost $c(s, a)$, and transitions to $s'$ under $\mathcal{P}(\cdot \mid s, a)$. For a policy $\pi$, we define the cumulative discounted value functions starting from $s$: $V_r^\pi(s) = \mathbb{E}[\sum_{t=0}^\infty \gamma^t r(s_t, a_t) \mid s_0 = s]$, $V_c^\pi(s) = \mathbb{E}[\sum_{t=0}^\infty \gamma^t c(s_t, a_t) \mid s_0 = s]$. The corresponding state-action value functions are

$$Q_r^\pi(s, a) = \mathbb{E}[\sum_{t=0}^\infty \gamma^t r(s_t, a_t) | s_0 = s, a_0 = a], \ Q_c^\pi(s, a) = \mathbb{E}[\sum_{t=0}^\infty \gamma^t c(s_t, a_t) | s_0 = s, a_0 = a]. \quad (1)$$

For discounted CMDPs, the optimal policy depends on the initial state (Sutton and Barto, 2018; Altman, 1999; Bertsekas, 2019). To make the analysis concrete and to address challenges in offline safe RL, we consider a fixed initial state $\rho = s$. The safe RL objective is to find a policy $\pi : \mathcal{S} \to \Delta(\mathcal{A})$ (where $\Delta(\cdot)$ denotes the probability simplex) that maximizes expected returns while satisfying a cost constraint:

$$\max_\pi \ \mathbb{E}_{s \sim \rho}[V_r^\pi(s)] \quad \text{s.t.} \quad \mathbb{E}_{s \sim \rho}[V_c^\pi(s)] \le l, \quad (2)$$

where $l$ is the maximum allowable cumulative discounted cost over trajectories generated by $\pi$.

**Offline Reinforcement Learning.** In online RL, a critical assumption is that the agent can interact with the environment almost indefinitely during training. As a result, estimation errors in the learned value functions can be corrected over time through online trial and error. In contrast, offline RL assumes access to a fixed dataset $\mathcal{D} = \{(s_i, a_i, r_i, c_i, s_i')\}_{i=1}^N$, consisting of $N$ transitions, including both safe and unsafe trajectories, collected by some behavior policy $\pi_\beta$. Most popular algorithms developed for online RL (Fujimoto et al., 2018; Haarnoja et al., 2018) cannot be directly applied in this setting. The key issue is that OOD actions $a'$ that were rarely or never visited by $\pi_\beta$ are often assigned unreliable and overly optimistic value estimates. When such actions are selected in the bootstrapped target under the Bellman operator: $\mathcal{T}^\pi Q_\diamond(s, a) := \diamond(s, a) + \gamma \mathbb{E}_{s' \sim \mathcal{P}, a' \sim \pi}[Q_\diamond(s', a')]$, where $\diamond := r$ or $c$, they can propagate and amplify

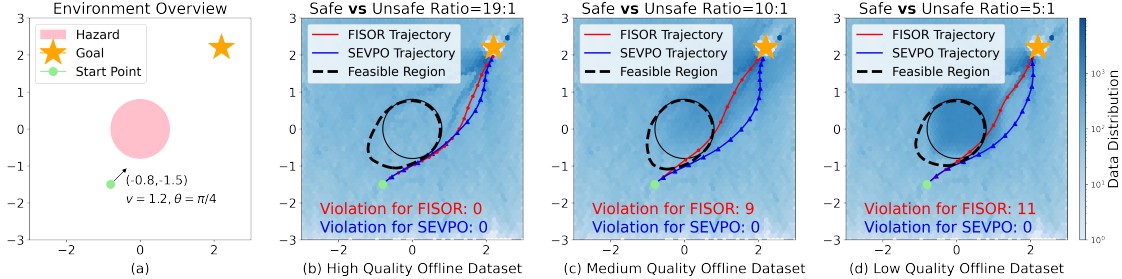

Figure 1: **Reach-avoid task:** The agent starts at $(-0.8, -1.5)$ (green) with velocity 1.2 and direction $\pi/4$, aiming to reach the goal (orange) while avoiding hazard zones (pink). Entering a hazard state incurs a cost of 1; other states have zero cost, and violations are counted as the number of steps spent in hazards. **(b)** High-quality data is collected from a TD3-Lag expert (Fujimoto et al., 2018). **(c)** Medium-quality data mixes expert and random policies (safe-to-unsafe 10 : 1). **(d)** More unsafe samples are added by increasing random trajectories in hazards (ratio 5 : 1). The "Feasible Region" denotes the largest safe region. Across all datasets, SEVPO learns a **safe**, optimal policy, while FISOR fails as data quality degrades.

extrapolation errors throughout the value function $Q_\diamond$. This challenge makes offline RL fundamentally different from the online setting, requiring careful control of distributional shift and support mismatch.

**Out of Distribution in Offline RL.** In actor–critic approaches, when the learned policy selects actions $a'$ that are rarely or never present in $\mathcal{D}$, the corresponding value estimates $Q(s', a')$ can be unreliable, leading to *extrapolation error* (Fujimoto et al., 2019) that propagates through the Bellman backups and destabilizes learning. This phenomenon makes it essential to design mechanisms that explicitly control distributional shift in both the critic and the actor.

**Value Function Learning (Critic Update).** The extrapolation error problem (Fujimoto et al., 2019) arises when estimating $Q_r(s', a')$ for actions $a'$ outside the dataset support. One widely adopted approach to mitigate this error is to restrict the value backup to actions that lie within the *support of the data distribution* (Kostrikov et al., 2022; Wu et al., 2019; 2022; Mao et al., 2023). Formally, these methods can be written as:

$$L(\theta) := \mathbb{E}_{(s,a,s')\sim\mathcal{D}}\Big[\big(Q_r(s,a;\theta) - r(s,a) - \gamma \max_{a'\in\mathcal{A}, \, \pi_\beta(a'|s')>0} Q_r(s', a';\hat{\theta})\big)^2\Big], \tag{3}$$

where $Q_r(s, a; \theta)$ is the parameterized value function, and $\hat{\theta}$ denotes a target network (e.g., updated via Polyak averaging). Another class of solutions constrains the learned policy to remain close to the behavior policy by introducing an explicit KL penalty during the Q-value update (Fujimoto and Gu, 2021; Peng et al., 2019; Nair et al., 2020; Wang et al., 2020):

$$L(\theta) := \mathbb{E}_{(s,a,r,s')\sim\mathcal{D}, \, a'\sim\pi_\phi(\cdot|s')}\Big[\big(Q_r(s,a;\theta) - r(s,a) - \gamma(Q_r(s', a';\hat{\theta}) - \alpha D_{\mathrm{KL}}[\pi_\theta(\cdot|s')\|\pi_\beta(\cdot|s')])\big)^2\Big],$$

where $D_{\mathrm{KL}}$ denotes the KL divergence between the current and behavior policies.

**Policy Improvement (Actor Update).** To further control distributional shift during policy improvement, the same divergence measure is often used as a regularization penalty for constrained actor optimization, a technique known as *policy regularization* (Peng et al., 2019; Peters and Schaal, 2007; Wang et al., 2018; Nair et al., 2020; Wu et al., 2022):

$$\max_{\pi_\phi} \mathbb{E}_{s\sim\mathcal{D}}\Big[\mathbb{E}_{a\sim\pi_\phi(\cdot|s)}[Q_r(s,a;\theta)] - \alpha D_{\mathrm{KL}}[\pi_\phi(\cdot|s)\|\pi_\beta(\cdot|s)]\Big].$$

These mechanisms form the basis of many state-of-the-art offline actor–critic algorithms by ensuring that both critic estimation and actor updates remain anchored to the dataset distribution, thereby mitigating extrapolation error and improving stability.

**Challenges and Limitations in safe RL.** Incorporating safety constraints into offline RL introduces challenges beyond standard distribution shift. The standard Q-learning update is not directly applicable (Wei et al., 2022b) because the value function must simultaneously capture both rewards and costs, which cannot

be handled by a single Bellman backup. A common approach is to formulate the problem as a constrained optimization and introduce a Lagrangian multiplier to balance reward and cost. In the offline setting, however, the dual variable cannot be reliably updated without rollouts, making it highly sensitive to value estimation errors.

Similar to offline RL, many safe RL algorithms consider solving:

$$\max_{\pi} \ \mathbb{E}_{s\sim\rho}[V_r^{\pi}(s)] \quad \text{s.t.} \ \mathbb{E}_{s\sim\rho}[V_c^{\pi}(s)] \leq l, \quad D(\pi\|\pi_{\beta}) \leq \epsilon, \tag{4}$$

where the divergence constraint $D(\pi\|\pi_{\beta})$ aims to prevent OOD actions (Lee et al., 2022; Xu et al., 2022; Lin et al., 2023; Zheng et al., 2024). However, this formulation faces several issues in the offline safe RL setting:

- **(i)** The optimization problem in Eq.(4) enforces the safety budget only in *expectation* over the initial state distribution. While this average-case guarantee is acceptable in benign reward-maximization tasks, it is inadequate for safety-critical applications: a policy can satisfy the constraint on aggregate while still producing individual states where the expected cumulative cost $V_c^{\pi}(s)$ exceeds the limit $l$.

- **(ii)** The constraints $V_c^{\pi}(s) \leq l$ and $D(\pi\|\pi_{\beta}) \leq \epsilon$ may be incompatible, leaving the optimization problem with no feasible solution. This conflict is exacerbated when multiple safety constraints or distinct cost functions are involved. In unconstrained offline RL, this issue does not arise because the optimization reduces to maximizing returns over policies near $\pi_{\beta}$ without enforcing cost constraints.

- **(iii)** Enforcing $D(\pi\|\pi_{\beta})$ uniformly across all states can force the learned policy to stay overly close to the behavior policy, even in regions where $\pi_{\beta}$ is unsafe or highly suboptimal. In offline safe RL this effect is particularly severe, as it makes the learned policy's performance strongly dependent on the quality of the dataset collected under $\pi_{\beta}$ (as illustrated in Figure 1).

Consider a simple MDP (Figure 2) with deterministic rewards and costs (discount factor is 1): states $s_1, s_3, s_5$ have reward $r = 1$ and cost $c = 0$, while $s_2$ and $s_4$ have reward $r = 1$ and cost $c = 1$. The behavior policy $\pi_{\beta}$ selects action $a_5$ in $s_2$, i.e., $\pi_{\beta}(s_2) = a_5$. The objective is to learn an optimal policy $\pi^*$ that reaches $s_5$ while minimizing cost violations. Using total variation as the divergence metric, we have $D(\pi\|\pi_{\beta}) = 1 - p$, where $p = \pi(a_5|s_2)$. First, if $s_2$ is the initial state, the standard optimization problem Eq.(4) has *no feasible solution* when $0 < \epsilon < 1$ and $l = 1$, because it simultaneously requires $p \geq 1 - \epsilon > 0$ and $V_c^{\pi}(s_2) = 1 + p \cdot V_c^{\pi}(s_4) = 1 + p > l$. Second, even if the cost limit $l$ is increased so that a feasible solution exists, setting $\epsilon$ sufficiently small forces the policy to choose $a_5$ with high probability, $p \geq 1 - \epsilon \approx 1$, causing the agent to never learn the optimal action $a_4$.

This example highlights the core difficulty: a uniform constraint $D(\pi\|\pi_{\beta}) \leq \epsilon$ can make the optimization problem infeasible or push the learned policy to simply mimic $\pi_{\beta}$, especially in states where the behavior policy is unsafe. Therefore, it is essential to distinguish between *"good" states*, where the policy can satisfy the constraints, and *"bad" states*, where violations are unavoidable, and to apply different training strategies to address infeasibility and poor data quality.

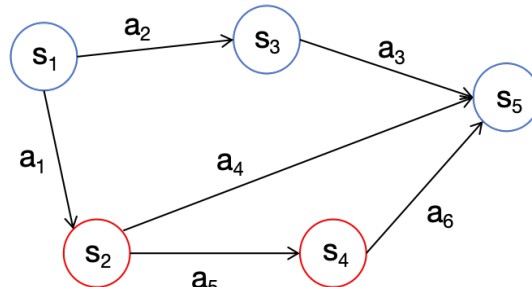

Figure 2: An MDP example

## 3 Selective Value and Policy Optimization

To address the limitations discussed earlier, we propose a *selective value learning and policy optimization* framework for actor–critic approaches in offline safe RL. The key idea is to decompose the original CMDP optimization problem Eq.(2) into subproblems and apply different critic and actor updates, with or without behavior regularization, based on the safety status of each state. We begin by partitioning the state space into a *safe region* $(\mathcal{S}_{\text{safe}}^{\pi})$ and an *unsafe region* $(\mathcal{S}_{\text{unsafe}}^{\pi})$, defined with respect to a given policy $\pi$:

**Definition 1** *The* **safe region** *under a policy $\pi$ is defined as $\mathcal{S}_{safe}^{\pi} := \{s \in \mathcal{S} \mid V_c^{\pi}(s) \leq l\}$, and the unsafe region is its complement: $\mathcal{S}_{unsafe}^{\pi} := \mathcal{S} \setminus \mathcal{S}_{safe}^{\pi}$.*

This decomposition allows us to solve subproblems in each region. In the safe region, the objective is to maximize cumulative reward while satisfying the cost constraint and maintaining alignment with the behavior policy (via behavior regularization). This corresponds exactly to the standard safe RL objective in Eq. (5), applied to states within $\mathcal{S}_{\text{safe}}$. In contrast, for states in the unsafe region, the focus shifts to minimizing cumulative cost—reward becomes secondary since any trajectory starting in these states may violate the safety budget.

However, accurately identifying $\mathcal{S}_{\text{safe}}^{\pi}$ is generally infeasible in offline RL, as it requires Monte Carlo estimation of $V_c^{\pi}(s)$ via interaction with the environment. To overcome this challenge, we instead define a conservative, policy-independent approximation:

**Definition 2** *The* **largest safe region** *is defined as $\mathcal{S}_{safe} := \{s \in \mathcal{S} \mid \check{V}_c(s) \leq l\}$, where $\check{V}_c(s) := \min_{\pi} V_c^{\pi}(s)$ is the cost value under the most conservative policy. The corresponding* smallest unsafe region *is $\mathcal{S}_{unsafe} := \mathcal{S} \setminus \mathcal{S}_{safe}$.*

For any state outside the largest safe region, **no policy** can guarantee safety. In such cases, the optimal strategy is to minimize cumulative cost, i.e., to exit the unsafe region as quickly as possible. This partitioning of the largest safe region offers several benefits: **(i)** It can be estimated reliably in the offline setting *prior to* policy training, with its accuracy depending solely on the offline dataset rather than the learned policy (see Section 4). **(ii)** For states outside the largest safe region, policy optimization reduces to a cost-minimization problem, allowing the use of standard methods for unconstrained RL. **(iii)** Excluding behavior regularization in unsafe regions prevents infeasibility due to conflicting constraints and enables more flexible and effective learning. Based on this, we consider the following optimization problem:

$$\textbf{If } s \in \mathcal{S}_{\text{safe}} : \max_{\pi} V_r^{\pi}(s) \qquad \textbf{If } s \in \mathcal{S}_{\text{unsafe}} : \max_{\pi} -V_c^{\pi}(s)$$

$$\text{s.t.} V_c^{\pi}(s) \leq l, \ \bar{D}(\pi||\pi_{\beta}) \leq \epsilon. \tag{5}$$

Here, $\bar{D}$ is a normalized divergence metric computed only over states in $\mathcal{S}_{\text{safe}}$ (see Appendix B). We thus constrain the optimized policy to remain close to the behavior policy *only for states* in the safe region, while omitting behavior regularization outside it.

**Remark 3** *Excluding behavior regularization for states in $\mathcal{S}_{unsafe}$ is essential. In offline RL, behavior regularization is commonly used to mitigate OOD actions. However, in our formulation, this is unnecessary because the cost value function in the unsafe region is learned via critic updates as in IQL (Kostrikov et al., 2022), which already controls OOD actions (see Section 4).*

In the following theorem, we show that the optimal solution to the decoupled optimization problem Eq.(5) dominates the solution of the traditional formulation Eq.(4):

**Theorem 4** *Assume a finite state and action space, and that the offline dataset provides sufficient coverage. For any state $s \in \mathcal{S}_{safe}$ (as defined in Definition 2), let $\pi_1^*$ denote the optimal solution to problem Eq.(4) and $\pi_2^*$ the optimal solution to our decoupled problem Eq.(5). Then, $V_r^{\pi_2^*}(s) \geq V_r^{\pi_1^*}(s)$, and $V_c^{\pi_2^*}(s) \leq l$. For any state $s \notin \mathcal{S}_{safe}$, the optimization problem Eq.(4)* **does not** *admit a feasible solution, and $V_c^{\pi_2^*}(s) \leq V_c^{\pi}(s)$ for any other policy $\pi$ under a fixed initial state $\rho = s$.*

These results naturally extend to the cases where $\rho = \Delta(\mathcal{S}_{\text{safe}})$ and $\rho = \Delta(\mathcal{S}_{\text{unsafe}})$. The detailed proof is provided in the Appendix. This extension follows by linearity since $V^{\pi}(\rho) = \mathbb{E}_{s \sim \rho}[V^{\pi}(s)]$.

## 4 Algorithm

In this section, we formally introduce the details of our algorithm SEVPO. Following the standard actor-critic approach, our algorithm includes four critic networks: $\check{Q}_c$ and $\check{V}_c$ to represent the cost $Q$-value function under the most **conservative** policy; $Q_r^{\pi}$ to represent the reward $Q$-value function under the current policy $\pi$; and $Q_c^{\pi}$ to represent the cost $Q$-value function under the current policy. We also use another actor neural network $z_{\theta}$ to represent the policy. Next, we introduce some key components of our algorithm (Alg. 1).

---
**Algorithm 1 SEVPO** (Selective Value Learning and Policy Optimization)

---
1: **Initialization:** offline dataset: $\mathcal{D}$, threshold: $\delta$, trust episode: $K_{\text{trust}}$, cost limit: $l$; Initialize networks: $\check{V}_c, \check{Q}_c, Q_r, Q_c, z_\theta$, hyper-parameters: $\mu_1, \mu_2, b_1, b_2, k$;
2: **Determine the largest feasible region:**
3: **for** each gradient step **do**
4:     Update $\check{V}_c$ using Eq.(6).
5:     Update $\check{Q}_c$ using Eq.(7).
6: **end for**
7: Safe Region:    $\mathcal{S}_1 = \{s | \check{V}_c(s) \leq l, \check{Q}_c(s,a) \leq l\}$.
8: Unsafe Region:    $\mathcal{S}_2 = \{s | \check{V}_c(s) > l\}$.
9: **for** each iteration $k$ **do**
10:    **Learning critic networks:**
11:        Update $Q_r$ using Eq.(8).
12:        Update $Q_c$ using Eq.(9).
13:    **Policy Update:**
14:        If $s \in \mathcal{S}_1$, calculate $\mathcal{L}_z^{\text{safe}}(\theta)$ (Eq.(13)).
15:        If $s \in \mathcal{S}_2$ :
16:            If $k \leq K_{\text{trust}}$, calculate $\mathcal{L}_z^{\text{unsafe}}(\theta)$ (Eq.(14)).
17:            If $k > K_{\text{trust}}$ calculate $\mathcal{L}_z^{\text{unsafe}}(\theta)$ (Eq.(15)).
18:        Calculate the policy loss using Eq.(16).
19:        Update policy $z_\theta$. Set $z' \leftarrow z_\theta$ every $k$ episode.
20: **end for**

---

## 4.1 Selective Value Function Learning

In the following, we introduce the selective critic network updates for states inside and outside the largest safe region.

**Learning the largest safe region $\mathcal{S}_{\text{safe}}$:** To estimate the largest safe region, we adopt IQL (Kostrikov et al., 2022) to learn the value function $\check{V}_c(s)$ and the $Q$-function $\check{Q}_c(s,a)$, where $\check{V}_c(s) = \min_a \check{Q}_c(s,a)$ is obtained via expectile regression[1]:

$$\mathcal{L}_{\check{V}_c} = \mathbb{E}_{(s,a)\sim\mathcal{D}} \left[ L^\tau \big( \check{Q}_c(s,a) - \check{V}_c(s) \big) \right], \tag{6}$$

$$\mathcal{L}_{\check{Q}_c} = \mathbb{E}_{(s,a,s')\sim\mathcal{D}} \left[ \big( c(s,a) + \gamma \check{V}_c(s') - \check{Q}_c(s,a) \big)^2 \right], \tag{7}$$

where $L^\tau(u) = |\tau - \mathbb{1}_{\{u>0\}}| u^2$, and $\mathbb{1}_{\{\cdot\}}$ is the indicator function. For $\tau \in (0.5, 1)$, this asymmetric loss assigns higher weight when $\check{Q}_c \leq \check{V}_c$. We select IQL for learning the largest safe region for two key reasons: (i) The expectile regression losses in Eq.(6) and Eq.(7) approximate a constrained value backup that remains within the support of the dataset, mitigating OOD issues in offline RL. (ii) Learning $\check{V}_c$ and $\check{Q}_c$ under the most conservative policy is independent of policy optimization, allowing us to obtain accurate estimates of $\check{V}_c$ and $\check{Q}_c$ *before* training the policy. The largest safe region is then determined using $\check{V}_c$ and the cost threshold $l$. In practice, we can also use the modified target $\left( (1-\alpha)c(s,a) + \alpha\big(c(s,a) + \gamma\check{V}_c(s')\big) - \check{Q}_c(s,a) \right)^2$ to handle cases where there is a small probability $1 - \alpha$ of transitioning to an absorbing state (emergency exit state in safe RL) (Bansal et al., 2017). The fixed point of this safety Bellman equation converges to the standard Bellman fixed point as $\alpha \to 1$.

**Update reward/cost $Q$ functions for $s \in \mathcal{S}_{\text{safe}}$:** For states in the largest safe region $\mathcal{S}_{\text{safe}}$, to solve the optimization problem as shown in Eq.(5), we need to estimate the cost and reward $Q-$value functions $Q_c^\pi$ and $Q_r^\pi$ under the current policy, thus a $Q-$ learning style critic update is not applicable due to the nature of the safe RL problem that the optimal policy is a stochastic policy (Altman, 1999). Therefore we update the critic networks in the largest safe region by minimizing the standard Bellman loss using the offline dataset:

---
[1]The sign differs from standard IQL because we solve a minimization problem.

$$\mathcal{L}_{Q_r^\pi} = \mathbb{E}_{(s,a,s')\sim\mathcal{D},\ a'\sim\pi_\theta}\left[\left(r(s,a)+\gamma\cdot Q_r^\pi(s',a')-Q_r^\pi(s,a)\right)^2\right] \tag{8}$$

$$\mathcal{L}_{Q_c^\pi} = \mathbb{E}_{(s,a,s')\sim\mathcal{D},\ a'\sim\pi_\theta}\left[\left(c(s,a)+\gamma\cdot Q_c^\pi(s',a')-Q_c^\pi(s,a)\right)^2\right] \tag{9}$$

**Remark 5** *The IQL estimates $\check{V}_c$ and $\check{Q}_c$ in Eq. (6)-Eq. (7) are intended to approximate the cost value functions under the **most conservative policy**, and are used **only** to identify the largest safe region $\mathcal{S}_{safe}$ via the threshold $l$. This is distinct from the cost minimization problem involving $Q_c^\pi$ that appears later in our policy learning stage. There, $Q_c^\pi$ is defined with respect to the* current *policy $\pi$, and we only require (and apply) it on states in the safe region (i.e., $s \in \mathcal{S}_{safe}$). Therefore, $\check{Q}_c$ and $Q_c^\pi$ serve different roles and should not be conflated.*

## 4.2 Selective Policy Optimization

As shown in the optimization problem Eq.(5), our practical algorithm adopts separated objectives. **For states in $\mathcal{S}_{\mathbf{safe}}$**, where $\check{V}_c(s) \leq l$, we maximize:

$$\frac{Q_r^\pi(s,a)}{Q_c^\pi(s,a)} \cdot \mathbb{1}_{\{Q_c^\pi(s,a)>\delta\}} + Q_r^\pi(s,a) \cdot \mathbb{1}_{\{Q_c^\pi(s,a)\leq\delta\}}, \tag{10}$$

where $a \sim \pi$ is sampled from the actor network $z_\theta$. This objective penalizes the reward $Q$-function based on the cost $Q$-value $Q_c^\pi(s,a)$ whenever $Q_c^\pi(s,a) > \delta$ for a hyperparameter $\delta$. Its structure serves two purposes: (i) to discourage potentially unsafe actions when the cost-$Q$ value exceeds a threshold; (ii) to mitigate critic estimation errors by introducing a soft constraint $\delta$ that controls when the reward $Q$ should be penalized. The intuition is that the safe and unsafe regions are defined using the value function under the most conservative policy. As a result, starting from a state in the safe region does not guarantee zero violations under the current policy $\pi$. Hence, the optimization must remain cautious. The selective policy optimization is therefore:

$$\textbf{If } s \in \mathcal{S}_{\text{safe}} : \max_{\pi,a\sim\pi} \mathbb{E}\left[A_r^\pi(s,a)\right]$$

$$\text{s.t.} \int_{\{a|\check{Q}_c(s,a)\leq l\}} \pi(a\mid s)da = 1,\ \bar{D}(\pi\|\pi_\beta) \leq \epsilon, \tag{11}$$

where $A_r^\pi(s,a) := \frac{Q_r^\pi(s,a)}{Q_c^\pi(s,a)} \cdot \mathbb{1}_{\{Q_c^\pi(s,a)>\delta\}} + Q_r^\pi(s,a) \cdot \mathbb{1}_{\{Q_c^\pi(s,a)\leq\delta\}}$.

**For states in $\mathcal{S}_{\mathbf{unsafe}}$**, we instead optimize:

$$\max_{\pi,a\sim\pi} \mathbb{E}\left[-\check{A}_c(s,a)\right] \quad \text{s.t.} \int_a \pi(a\mid s)da = 1, \tag{12}$$

where $\check{A}_c(s,a) = \check{Q}_c(s,a) - \check{V}_c(s)$. The resulting algorithm, SEVPO, promotes actions that yield cost-aware high rewards in the safe region while keeping the policy close to the behavior distribution. In the unsafe region, SEVPO prioritizes minimizing the conservative cost advantage, effectively searching for the "least-cost escape path."

**Diffusion Policy Update:** In SEVPO, we parameterize the policy network $\pi_\theta$ using a diffusion model $z_\theta$ following Zheng et al. (2024). We adopt a weighted actor loss (Zheng et al., 2024; Hansen-Estruch et al., 2023; Kang et al., 2024), but our formulation is selective.

Specifically, policy optimization in the **safe region** is performed by minimizing the diffusion training loss of the behavior policy $\pi_\beta$:

$$\min_\theta \mathbb{E}_{t,(s,a),z}\left[\|z - z_\theta(a_t,s,t)\|_2^2\right],$$

augmented with the weight function $w_{\text{safe}}(s,a) = \exp\left(\mu_1 A_r^\pi(s,a)\right) \cdot \mathbb{1}_{\{\check{Q}_c(s,a)\leq l\}}$, yielding the selective objective:

$$\min_\theta \mathcal{L}_z^{\text{safe}}(\theta) := \min_\theta \mathbb{E}_{t\sim\mathcal{U}([0,T])\ z\sim\mathcal{N}(0,I)\ (s,a)\sim D}\left[w_{\text{safe}}(s,a) \cdot \|z - z_\theta(a_t,s,t)\|_2^2\right], \tag{13}$$

where $a_t = \alpha_t a + \sigma_t z$ is the noisy action following the forward diffusion distribution $\mathcal{N}(a_t \mid \alpha_t a, \sigma_t^2 I)$, and $\alpha_t, \sigma_t$ are the noise schedules. For the **unsafe region**, where distributional shift constraints are removed, the policy is optimized by:

$$\min_\theta \mathcal{L}_z^{\text{unsafe}}(\theta) := \min_\theta \mathbb{E}_{(s,a\sim\pi_\theta)}\big[\exp(-\mu_2 \check{A}_c(s,a))\big], \tag{14}$$

where $a \sim \pi_\theta$ is sampled after solving the diffusion ODEs/SDEs (Song et al., 2020).

**Remark 6** *To further enhance performance, we incorporate an additional trust-region term into $\mathcal{L}_z^{unsafe}(\theta)$ after $K_{trust}$ training steps:*

$$\min_\theta \mathcal{L}_z^{unsafe}(\theta) := \min_\theta \mathbb{E}_{(s,a\sim\pi_\theta)}\Big[\exp(-\mu_2 \check{A}_c(s,a)) \cdot \|z_\theta(a_t,s,t) - z'(a_t,s,t)\|_2^2\Big], \tag{15}$$

*where $z'$ is a copy of the policy network corresponding to $\pi_{t-k}$, with $k$ controlling the update frequency. We set $K_{trust}$ to half the total training steps. This regularization constrains successive policy updates, similar to the trust-region mechanism in iTRPO Zhang and Tan (2024), while remaining consistent with the optimization problem in Eq.(5). Note that $z'$ **is not** the behavior policy $\pi_\beta$.*

Combining the two objectives yields the final weighted loss:

$$\min_\theta \mathcal{L}_z(\theta) := \min_\theta \big(b_1 \cdot \mathcal{L}_z^{\text{safe}}(\theta) + b_2 \cdot \mathcal{L}_z^{\text{unsafe}}(\theta)\big), \tag{16}$$

where $b_1, b_2 > 0$ control the trade-off between the safe and unsafe region objectives.

**Relationship to FISOR.** Our method is most closely related to FISOR(Zheng et al., 2024) in that both adopt a diffusion-based policy backbone for offline safe RL.

**Similarities.** Beyond the backbone, SEVPO shares several components with FISOR: (i) both identify safe/unsafe regions via conservative cost value estimation and thresholding; (ii) both employ region-wise objectives with distinct optimization goals in the safe versus unsafe regions; (iii) our definitions of the $w_{\text{safe}}$(in Eq.(13)) and $\exp(-\mu_2 \check{A}_c(s,a))$(in Eq.(14)) follow FISOR.

However, SEVPO differs from FISOR in both *problem focus* and *algorithmic design*:

- **Distinct Problem focus.** FISOR targets improving performance on standard offline safe RL benchmarks under a unified training objective. In contrast, SEVPO is explicitly designed to address a practical robustness failure mode: learning from *mixed-quality* offline datasets that contain conflicting expert and low-quality/unsafe trajectories. In such settings, applying a uniform behavior-regularized update across all states can undesirably propagate unsafe behaviors, especially in states where maintaining safety is intrinsically difficult.

- **Algorithmic Innovations.** Beyond the shared diffusion backbone, SEVPO introduces a region-aware training mechanism based on safe/unsafe state partitions. Unlike FISOR, which applies a global optimization objective and behavior regularization uniformly, SEVPO conditions policy optimization on the estimated region:

  - **Region-aware behavior regularization:** FISOR enforces behavior regularization uniformly. In contrast, SEVPO applies a region-wise constraint (Eq.(5)). We enforce regularization *only* in $\mathcal{S}_{\text{safe}}$ to stabilize reward maximization, but explicitly *remove* it in $\mathcal{S}_{\text{unsafe}}$. This ensures the policy is not forced to imitate unsafe behaviors when the data quality is poor.
  - **Selective objectives and stabilization:** we use a conservative selective update in $\mathcal{S}_{\text{safe}}$ (Eq.(10)) and a recovery-oriented objective in $\mathcal{S}_{\text{unsafe}}$ (Eq.(12)), together with a trust-style stabilization mechanism (Eq.(15)).

These design choices directly target the mixed-quality failure mode and are not present in FISOR.

Table 1: Normalized DSRL benchmark results with Data Vaires. ↑ means the higher the better. ↓ means the lower the better. Each value is averaged over 20 evaluation episodes. Gray: Unsafe agents. **Bold**: Safe agents whose normalized cost is smaller than 1. **Blue**: Safe agents with the highest reward.

| Methods | Data Ratio | | CarButton1 | | CarButton2 | | CarGoal1 | | CarGoal2 | | CarPush1 | | CarPush2 | | AntVelocity | | HalfCheetah | | SwimmerVel | | Average | |
|---|---|---|---|---|---|---|---|---|---|---|---|---|---|---|---|---|---|---|---|---|---|---|
| | Safe | Unsafe | r↑ | c↓ | r↑ | c↓ | r↑ | c↓ | r↑ | c↓ | r↑ | c↓ | r↑ | c↓ | r↑ | c↓ | r↑ | c↓ | r↑ | c↓ | r↑ | c↓ |
| CPQ | ori | ori | 0.49 | 26.65 | 0.20 | 32.59 | 0.60 | 4.87 | 0.18 | 4.09 | 0.10 | 5.21 | 0.07 | 13.76 | -1.01 | 0 | 0.675 | 14.85 | 0.001 | 0 | 0.145 | 11.3 |
| | 8 | 1 | 0.25 | 46.3 | 0.21 | 50.8 | 0.21 | 3.05 | 0.001 | 6.84 | 0.28 | 1.21 | -0.04 | 17.395 | -1.012 | 0 | 0.019 | 0.215 | 0.002 | 0 | -0.008 | 13.97 |
| | 1 | 1 | 0.03 | 22.3 | 0.04 | 39.40 | 0.25 | 5.92 | 0.004 | 4.97 | -0.02 | 0.085 | 0.04 | 9.95 | -1.011 | 0 | -0.19 | 0 | 0.03 | 1.35 | -0.09 | 9.33 |
| | 1 | 3 | 0.10 | 44.4 | 0.11 | 27.01 | 0.59 | 3.995 | 0.24 | 20.07 | -0.01 | 7.49 | 0.04 | 34.88 | -1.011 | 0 | -0.25 | 0 | 0.03 | 1.02 | -0.017 | 15.42 |
| COptiDICE | ori | ori | -0.09 | 4.41 | -0.02 | 3.76 | 0.47 | 1.68 | 0.12 | 1.76 | 0.10 | 5.21 | 0.23 | 1.54 | 0.16 | 9.38 | 0.61 | 0 | 0.7 | 36.3 | 0.35 | 7.84 |
| | 8 | 1 | 0.04 | 2.7 | -0.02 | 0.93 | 0.36 | 3 | 0.24 | 3.23 | 0.23 | 2.165 | 0.05 | 2.42 | 0.96 | 3.285 | 0.51 | 0 | 0.48 | 5.4 | 0.316 | 2.57 |
| | 1 | 1 | -0.09 | 0.86 | -0.12 | 3.65 | 0.26 | 1.48 | 0.14 | 2.16 | 0.22 | 1.26 | 0.11 | 4.91 | 0.99 | 13.99 | 0.50 | 0 | 0.50 | 15.1 | 0.27 | 4.82 |
| | 1 | 3 | -0.23 | 1.38 | -0.29 | 3.54 | 0.22 | 2.01 | 0.08 | 1.97 | 0.25 | 3.60 | 0.15 | 4.99 | 0.97 | 11.62 | 0.46 | 0 | -0.05 | 1.465 | 0.17 | 3.39 |
| FISOR | ori | ori | -0.11 | 0.73 | -0.001 | 0.83 | 0.42 | 1.63 | 0.03 | 0.24 | 0.24 | 1.61 | 0.10 | 1.21 | 0.89 | 0.003 | 0.89 | 0 | -0.04 | 0.31 | 0.268 | 0.72 |
| | 8 | 1 | 0.01 | 1.12 | 0.0083 | 1.23 | 0.458 | 1.505 | 0.03 | 0.355 | 0.27 | 0.54 | 0.07 | 0.24 | 0.88 | 0 | 0.87 | 0 | -0.06 | 0.125 | 0.28 | 0.56 |
| | 1 | 1 | 0.01 | 1.045 | -0.02 | 1.75 | 0.165 | 0.75 | 0.06 | 2.49 | 0.27 | 2.12 | 0.08 | 1.82 | 0.78 | 0 | 0.83 | 0 | 0.312 | 1.24 | 0.26 | 1.43 |
| | 1 | 3 | -0.09 | 1.455 | -0.006 | 2.655 | 0.183 | 1.505 | 0.14 | 2.08 | 0.20 | 1.67 | -0.01 | 1.355 | 0.77 | 0 | 0.84 | 0 | 0.33 | 2.185 | 0.27 | 1.24 |
| CCAC | ori | ori | 0.42 | 37.65 | 0.34 | 43.03 | -0.31 | 0.64 | -0.96 | 4.51 | -1.69 | 0.43 | 0.0013 | 0 | -0.89 | 0 | -0.17 | 0 | -0.006 | 1.13 | -0.36 | 9.71 |
| | 8 | 1 | 0.11 | 30.63 | -1.00 | 22.93 | 0.43 | 10 | 0.60 | 22.02 | -0.35 | 1.56 | 0.05 | 20.09 | 0.27 | 0 | -0.25 | 1.33 | -0.08 | 0.03 | 0.03 | 12.06 |
| | 1 | 1 | -0.0003 | 19.75 | 0.0003 | 27.82 | 0.78 | 6.07 | 0.90 | 19.765 | 0.21 | 3.625 | 0.14 | 25.05 | -1.01 | 0 | -0.27 | 1.275 | -0.07 | 0.025 | 0.07 | 11.48 |
| | 1 | 3 | 0.30 | 48.32 | 0.31 | 46.9 | 0.78 | 6.21 | 0.89 | 19.75 | 0.18 | 2.93 | 0.11 | 27.47 | -1.012 | 0 | -0.25 | 1.23 | -0.08 | 0 | 0.13 | 16.97 |
| SEVPO | ori | ori | -0.04 | 0.51 | -0.04 | 0.27 | 0.38 | 0.48 | 0.07 | 0.44 | 0.18 | 0.47 | 0.148 | 0.61 | 0.928 | 0.211 | 0.966 | 0.08 | -0.05 | 0.125 | 0.282 | 0.35 |
| | 8 | 1 | -0.15 | 0.535 | -0.10 | 0.34 | 0.35 | 0.52 | 0.03 | 0.505 | 0.18 | 0.53 | 0.12 | 0.56 | 0.96 | 0.32 | 0.88 | 0.005 | -0.05 | 0.86 | 0.24 | 0.46 |
| | 1 | 1 | -0.07 | 0.565 | -0.08 | 0.63 | 0.21 | 0.16 | 0.002 | 0.71 | 0.11 | 0.54 | 0.01 | 0.41 | 0.81 | 0.045 | 0.88 | 0 | -0.03 | 0.54 | 0.20 | 0.39 |
| | 1 | 3 | -0.25 | 0.1 | -0.25 | 0.56 | 0.08 | 0.21 | 0.001 | 0.685 | 0.10 | 0.43 | 0.004 | 0.49 | 0.71 | 0.047 | 0.86 | 0.005 | -0.04 | 0.075 | 0.136 | 0.28 |

Table 2: Normalized results for the impact of policy restriction

| Task | w/o policy restriction | | with policy restriction | |
|---|---|---|---|---|
| | reward ↑ | cost ↓ | reward ↑ | cost ↓ |
| SwimmerVel | -0.06 | 0.02 | -0.05 | 0.125 |
| AntVel | 0.75 | 0.01 | 0.928 | 0.211 |
| HalfCheetah | 0.36 | 0 | 0.966 | 0.08 |

## 5 Experiments

**Evaluation Setups.** We evaluate SEVPO on Safety-Gymnasium (Ray et al., 2019; Ji et al., 2023) tasks from the DSRL benchmark (Liu et al., 2023a) to compare against state-of-the-art offline safe RL algorithms. To analyze performance under varying data quality, a setting rarely explored in safe offline RL, we construct three scenarios: high-quality datasets with predominantly safe data (safe-to-unsafe $8:1$), medium-quality datasets with balanced data $(1:1)$, and low-quality datasets with predominantly unsafe data $(1:3)$. These distributions are generated by selectively sampling safe trajectories and duplicating unsafe ones. We report *normalized return* and *normalized cost* as evaluation metrics, where a normalized cost below 1 indicates safety. Following DSRL principles, safety is treated as the primary objective, with the cost limit for all Safety-Gymnasium tasks set to 10 to better simulate safety-critical conditions.

**Baselines.** We compare our algorithm with the following baselines: CPQ(Xu et al., 2022): Designates actions outside the data distribution as unsafe and updates the reward critic with safe state-action pairs; COptiDICE(Lee et al. (2022)): Estimates the stationary distribution corrections for the optimal policy in terms of returns with an upper bound on costs; FISOR(Zheng et al. (2024)): Learns the feasible region to maximize rewards within it while minimizing constraint violations in the unfeasible region. CCAC(Guo et al. (2025)): Learns constraint-aware policies for varying budgets using a Constrained VAE model.

**Simulation Results.** To highlight the effectiveness of our approach, Table 1 reports performance on the original expert-collected dataset (Liu et al., 2023a) as well as under varying dataset distributions. On the original dataset, SEVPO consistently learns a safe policy across all tasks and achieves the highest rewards in most environments, performing on par with the state-of-the-art algorithm FISOR. More importantly, as the safe-to-unsafe data ratio shifts, even slightly, none of the existing state-of-the-art methods can reliably maintain a safe policy across all tasks. In contrast, SEVPO is the **only** algorithm that consistently preserves a **safe** policy under every data distribution while simultaneously achieving the highest overall rewards in all scenarios. These results demonstrate SEVPO's robustness to dataset quality and its ability to adapt to environments where the proportion of safe and unsafe data varies significantly. They also validate our analysis of the core challenges in offline safe RL and underscore the importance of a behavior-aware, region-based learning framework.

**Worst-Case Risk and Failure Frequency** In addition to normalized reward and expected cost, we further evaluate safety-critical metrics that explicitly characterize worst-case risk and failure frequency. Specifically,

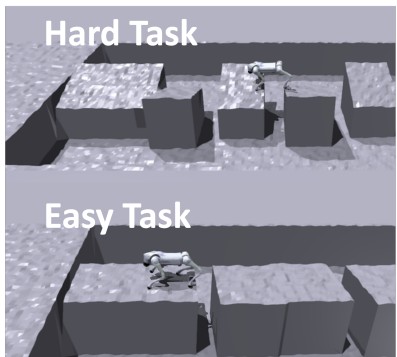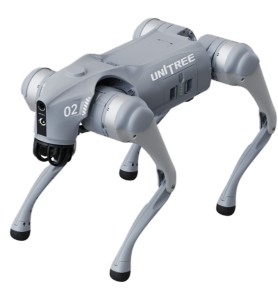

Figure 3: Navigation Task Overview

Table 3: Worst-case safety metrics across six tasks.

| | FISOR | | SEVPO | |
|---|---|---|---|---|
| Task | Max Episode Cost ↓ | Failure Rate ↓ | Max Episode Cost ↓ | Failure Rate ↓ |
| CarPush2 | 152 | 70% | **37** | **10%** |
| CarPush1 | 73 | 73% | **30** | **17%** |
| CarButton2 | 85 | 65% | **61** | **12%** |
| CarButton1 | 154 | 95% | **58** | **25%** |
| CarGoal2 | 79 | 65% | **62** | **40%** |
| CarGoal1 | 108 | 70% | **30** | **17%** |

we report: (i) the *maximum per-episode cost* across evaluation rollouts, which captures worst-case risk, and (ii) the *failure rate*, defined as the fraction of episodes whose total cost exceeds a fixed cost limit of $l = 10$. All results are computed over 60 evaluation episodes using policies trained on **low-quality** data. Table 3 summarizes the worst-case safety metrics on six Safety-Gymnasium tasks. Across all tasks, SEVPO consistently reduces the maximum per-episode cost relative to the baseline FISOR, indicating improved robustness in worst-case safety outcomes. Moreover, SEVPO achieves substantially lower failure rates on five out of six tasks, with particularly large gains on CarPush2 ($70\% \rightarrow 10\%$) and CarButton1 ($95\% \rightarrow 25\%$). These results demonstrate that SEVPO improves safety stability beyond average performance, by mitigating extreme-cost episodes and reducing the frequency of safety violations.

**Performance under Different Cost Limit.** To examine whether SEVPO is sensitive to the choice of the cost limit $l$, we conduct an ablation study on CarGoal2 with $\delta \in \{10, 20, 40\}$. As shown in Table 4, SEVPO consistently maintains low costs across different limits while achieving improved rewards as the constraint is relaxed, indicating that its performance is not driven by a particular choice of $l$.

Table 4: Sensitivity analysis of SEVPO under varying cost limits on CarGoal2.

| | Cost Limit $\delta = 10$ | | Cost Limit $\delta = 20$ | | Cost Limit $\delta = 40$ | |
|---|---|---|---|---|---|---|
| Method | Reward ↑ | Cost ↓ | Reward ↑ | Cost ↓ | Reward ↑ | Cost ↓ |
| SEVPO | 0.07 | 0.44 | 0.10 | 0.24 | 0.20 | 0.75 |

**Safe Offline RL with New Data Collection.** To examine the effect of data quality and test SEVPO's adaptability, we design a setting where the agent can collect a *new dataset* during training. This mimics a practical scenario where a policy is first trained on a low-quality dataset and then gathers additional data once it becomes **safe enough** to explore. Starting from a 1 : 1 safe-to-unsafe ratio, SEVPO is trained for $2 \times 10^5$ steps, collects new data, continues training for another $2 \times 10^5$, and performs a second data collection. After the first collection, the mean episode cost drops from 69.72 to 7.45 (Figure 4(c)), indicating the policy is already safe for exploration. As shown in Figure 4(a,b), incorporating new data boosts rewards by **72%** after $4 \times 10^5$ steps and **26%** after $6 \times 10^5$, with all policies remaining safe. Additional details are provided in Section D.

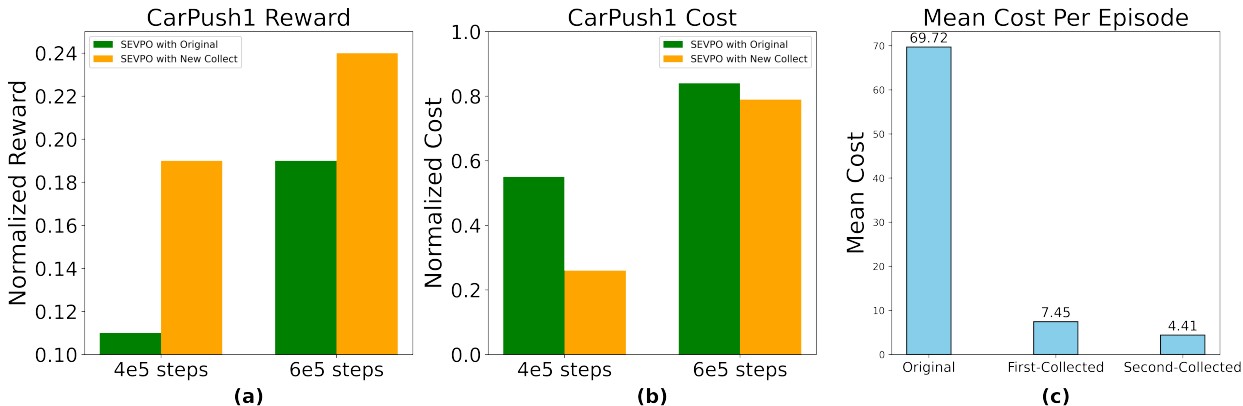

Figure 4: (a), (b): Normalized reward and cost during training on CarPush 1; (c): The average cost per episode (cost limit $l$ = 10).

**Ablation Study on Updating Policy for States in the Unsafe Region.** As discussed in Section 1, introducing an additional restriction on the trust region when updating the policy for states in the unsafe region prevents the policy from deviating excessively. We evaluate SEVPO's performance with and without the trust-region term $z'(a_t, s, t)$ defined in Eq.(15). The results in Table 4.2 show that incorporating this policy restriction improves the achieved reward, while SEVPO still maintains safety even without this regularization.

**Real world Quadruped Navigation Application** To further evaluate the practicality of SEVPO, we conduct experiments in Nvidia Isaac Gym (Makoviychuk et al., 2021) with the Unitree Go2 quadruped robot. We design two dynamic, safety-critical navigation tasks (Fig. 3) to test the algorithm's ability to navigate and recover under realistic conditions. Following a mixed-quality offline-data regime, the training dataset consists of 6,000 expert trajectories and 3,000 failure ("fall-down") trajectories, which mimics imperfect real-world data collection. For quantitative evaluation, we run 20 independent test episodes and report the success and fall-down rates in Table 5. A demonstration video of both tasks is available at `https://youtu.be/tDpWq2EV_Ig`.

Table 5: Quantitative comparison on Quadruped Navigation

| Metric | FISOR (Baseline) | SEVPO (Ours) |
|---|---|---|
| **Success Rate** ↑ | 5% | **85%** |
| **Fall-down Rate** ↓ | 95% | **15%** |

As shown in Table 5, SEVPO dramatically improves task completion under mixed-quality data, achieving an 85% success rate (vs. 5% for FISOR) and reducing the fall-down rate from 95% to 15%. Note that a fall-down is a terminal failure that typically occurs early in an episode (e.g., at the first or second gap), so success/failure statistics provide the most direct and informative measure of reliability in this setting. These results demonstrate that SEVPO is robust to failure trajectories in the offline dataset and can maintain strong real-world navigation performance.

# 6 Conclusion

In this work, we proposed SEVPO, a divide-and-conquer framework for offline safe RL that performs selective value learning and policy optimization based on state safety. SEVPO learns conservative cost values to reliably identify safe states, applies reward-constrained optimization with selective regularization in those regions, and switches to cost minimization elsewhere to compute least-cost escape paths. Extensive experiments across diverse dataset qualities show that SEVPO achieves high reward while enforcing strict safety guarantees, consistently outperforming state-of-the-art offline safe RL methods, and our Unitree Go2 experiments in dynamic environments trained purely from offline data highlight its potential for real-world safety-critical robotics.

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

# Appendix

# A  More Related Works

Safe RL has attracted great attention recently, both in online setting (Efroni et al., 2020; Liu et al., 2021; Bura et al., 2022; Wei et al., 2022b;a; 2024a; Müller et al., 2023; Germano et al., 2023; Ding et al., 2024; Müller et al., 2024) and offline setting (Wu et al., 2021; Chen et al., 2021; Lee et al., 2022; Liu et al., 2023c; Xu et al., 2022; Guan et al., 2024; Zheng et al., 2024; Hong and Tewari, 2024).

In theoretical offline RL, certain assumptions about the ratio between the state-action occupancy distribution of a policy $\pi$ and the dataset distribution $\mu$ are necessary to prove convergence. This ratio is represented by $w^\pi = d^\pi/\mu$. A commonly used assumption is that the $\ell_\infty$ concentrability $C^\pi_{\ell_\infty}$, defined as the infinite norm of $w^\pi$, is bounded for all policies (Liu et al., 2019; Chen and Jiang, 2019; Wang et al., 2019; Liao et al., 2022; Zhang et al., 2020). However, this assumption is difficult to satisfy, particularly in offline safe RL, as it requires the dataset to adequately cover all unsafe state-action pairs. To address this, recent works (Rashidinejad et al., 2021; Zhan et al., 2022; Chen and Jiang, 2022; Xie et al., 2021; Uehara and Sun, 2021) introduce pessimism to reduce the requirement to only the best-covered policy. Zhu et al. (2023) further relaxes this assumption by introducing $\ell_2$ concentrability, which is always bounded by $\ell_\infty$. While existing offline safe RL studies (Hong et al., 2024; Le et al., 2019) still require coverage for all policies, recent works Wei et al. (2024b); Zhang et al. (2024) relax this assumption to single-policy coverage.

Deep offline safe RL algorithms (Lee et al., 2022; Liu et al., 2023c; Xu et al., 2022; Chen et al., 2021; Zheng et al., 2024; Chemingui et al., 2026) have shown strong empirical performance but usually lack theoretical guarantees. As we mentioned in the introduction, balancing generalization with preventing undesirable behaviors from out-of-distribution actions is a key challenge in offline RL. A straightforward approach is to directly constrain the learned policy to align with the behavior policy used to collect the data set (Fujimoto et al., 2019; Kumar et al., 2019; Fujimoto and Gu, 2021; Wu et al., 2019). Other methods address this by making conservative estimates of future rewards through a lower bound on the true value function (Kumar et al., 2020; Yu et al., 2021) or by penalizing out-of-distribution (OOD) actions to control distributional shift (Kostrikov et al., 2021; Lyu et al., 2022). Model-based approaches often handle uncertainty by using ensembles, penalizing actions with high variability across models and promoting those that show consistency (Janner et al., 2019; Kidambi et al., 2020). Recently, some studies have explored the use of expressive diffusion models for policy learning to capture more complex distributions, achieving impressive results (Janner et al., 2022; Lu et al., 2023; Hansen-Estruch et al., 2023; Zheng et al., 2024).

## A.1  Discussions on the Comparison between SEVPO and FISOR

Our formulation is inspired by FISOR (Zheng et al., 2024), which proposes a feasibility-guided diffusion model for offline safe RL. However, our formulation differs from FISOR in several significant ways:

- In terms of formulation, FISOR addresses only the "hard constraint" setting, where the cost value function represents the maximum constraint violation (largest state-wise cost) in the trajectory. This greatly limits the generalization of their approach in practice. In contrast, SEVPO can handle a wide range of constraints, offering flexibility by adjusting the constraint limit $l$.

- We remove behavior cloning for states in the largest unsafe region, which is crucial for learning a safe policy, especially when the behavior policy performs poorly in these states. To the best of our knowledge, this formulation and the accompanying theoretical results are unique in the offline safe RL community.

- FISOR defines the feasible region using the value function for the hard constraint based on Hamilton-Jacobi (HJ) reachability (Bansal et al., 2017) from control theory. However, this formulation applies only to **deterministic systems**, which is rarely the case in most RL environments, and cannot be generalized to problems with more flexible cost limits. The Bellman equation under their value function no longer holds in stochastic systems.

- From a practical algorithm design perspective, we incorporate several key components to better stabilize learning and improve performance.

### A.2 Discussions recent work on data-quality and OOD generalization

Recently, beyond offline RL, there has also been growing interest in understanding robustness issues from a *data-centric* and OOD generalization perspective. For graph anomaly detection with few labels, Ma et al. (2024) leverage denoising diffusion models to synthesize auxiliary labeled nodes consistent with the original attributes, improving existing detectors without changing their architectures. For streaming data, Yu et al. (2023) view concept drift as data-quality shift and propose a lightweight type-driven drift detector pre-trained on synthetic streams and adapted via knowledge distillation to generalize under evolving distributions. From an OOD view of adversarial shifts on graphs, Li et al. (2024) treat poisoning as OOD samples and incorporate OOD detection into adversarial training to improve robustness. However, these approaches are tailored to supervised or streaming graph settings and are difficult to directly transfer to RL.

## B   Proof of Theorem 4

For any state $s \in \mathcal{S}$, with $\rho = s$, define the standard optimization problem as:

$$\max_{\pi} V_r^{\pi}(s) \quad \text{s.t.} \quad V_c^{\pi}(s) \le l, \ D(\pi\|\pi_\beta) \le \epsilon, \tag{17}$$

In the offline safe RL setting, we define $D(\pi\|\pi_\beta) := \sum_{s\in\mathcal{S}} \sum_{a\in\mathcal{A}} \mu(s)\pi(a|s) \log\left(\frac{\pi(a|s)}{\pi_\beta(a|s)}\right)$, where $\mu(s)$ is the marginal distribution of state under the behavior policy $\pi_\beta$. The sufficient coverage over a policy $\pi$ will ensure that $\pi(a|s) \log \frac{\pi(a|s)}{\pi_\beta(a|s)} > 0$.

Our decoupled behavior-aware, region-based formulation is:

$$\text{If} \quad s \in \mathcal{S}_{\text{safe}} : \max_{\pi} V_r^{\pi}(s); \qquad\qquad\qquad \text{If} \quad s \in \mathcal{S}_{\text{unsafe}} : \max_{\pi}(-V_c^{\pi}(s))$$
$$\text{s.t.} \quad V_c^{\pi}(s) \le l, \bar{D}(\pi\|\pi_\beta) \le \epsilon, \tag{18}$$

where we consider the KL divergence as the normalized distance metric $\bar{D}$, which is defined as: $\bar{D} := \sum_{s\in\mathcal{S}_{\text{safe}}} \sum_{a\in\mathcal{A}} \mu(s)\pi(a|s) \log\left(\frac{\pi(a|s)}{\pi_\beta(a|s)}\right)$. Let $\pi_1^*$ be the optimal solution of the optimization problem equation 17, and $\pi_2^*$ be the optimal solution of the decoupled optimization problem equation 18. To show that $\pi_2^*$ dominates $\pi_1^*$, we will show that $\pi_1^*$ is a feasible solution to the optimization problem equation 18. First, for any $s \in \mathcal{S}_{\text{safe}}$, we have that $V_c^{\pi_1^*}(s) \le l, D(\pi_1^*\|\pi_\beta) \le \epsilon$ according to the constraints in equation 17. Since we have

$$\begin{aligned}
\epsilon \ge D(\pi_1^*\|\pi_\beta) &= \sum_{s\in\mathcal{S}} \sum_{a\in\mathcal{A}} \mu(s)\pi_1^*(a|s) \log\left(\frac{\pi_1^*(a|s)}{\pi_\beta(a|s)}\right) \\
&= \sum_{s\in\mathcal{S}_{\text{safe}}} \sum_{a\in\mathcal{A}} \mu(s)\pi_1^*(a|s) \log\left(\frac{\pi_1^*(a|s)}{\pi_\beta(a|s)}\right) + \sum_{s\in\mathcal{S}_{\text{unsafe}}} \sum_{a\in\mathcal{A}} \mu(s)\pi_1^*(a|s) \log\left(\frac{\pi_1^*(a|s)}{\pi_\beta(a|s)}\right) \\
&\ge \sum_{s\in\mathcal{S}_{\text{safe}}} \sum_{a\in\mathcal{A}} \mu(s)\pi_1^*(a|s) \log\left(\frac{\pi_1^*(a|s)}{\pi_\beta(a|s)}\right).
\end{aligned}$$

Thus, $\pi_1^*$ is also a feasible solution to the optimization problem equation 18 as well. Therefore we have $V_r^{\pi_2^*}(s) \ge V_r^{\pi_1^*}(s)$. For the case when $s \notin \mathcal{S}_{\text{safe}}$, the optimization problem equation 17 **does not** have a solution. $V_c^{\pi_2^*}(s)$ is the **smallest** cumulative cost we can have.

## C   Experimental Details

### C.1   Reach-avoid Task

The experimental setup illustrated in Figure 1, as adapted from the toy case experiment discussed in (Zheng et al., 2024), entails an agent tasked with navigating towards a goal while circumventing a hazard located

at the center of the map. The state space of the agent is defined as $S := (x, y, v, \theta)$, where $x$ and $y$ are the agent's coordinates, $v$ represents the initial velocity, and $\theta$ the direction of movement. The action space is designated as $A := (\bar{v}, \bar{\theta})$, with $\bar{v}$ denoting acceleration and $\bar{\theta}$ angular acceleration. The reward function $r$ evaluates performance by measuring the change in distance to the target between consecutive time steps. To encourage the agent to reach the goal location, we assign a substantial reward when the agent successfully arrives at the goal. The cost function $h$ is expressed as:

$$h := R_{\text{hazard}} - d_{\text{hazard}}$$

In this expression, $R_{\text{hazard}}$ represents the radius of the hazard, set to 0.8 in this implementation, and $d_{\text{hazard}}$ denotes the distance between the agent and the hazard. A negative or zero value of $h$ indicates a collision with the hazard, signaling a breach of safety. The compliance measure $c$ is then defined as $c := \mathbb{1}_{\{h \leq 0\}}$, which assesses the agent's adherence to safety protocols during navigation.

To gather sufficient data, we first allow the agent to interact with the showcase environment and train it using the TD3-Lag algorithm, as modified by (Fujimoto et al., 2018). Once the agent is well-trained, it is deployed to collect expert data. Additionally, we run a random policy to accumulate a large number of unsafe or low-quality data samples. These random policy samples are then combined with the expert data to generate three distinct categories of offline datasets: high-quality, with a safe-to-unsafe ratio of 19:1; medium-quality, with a ratio of 10:1; and low-quality, with a ratio of 5:1. The distribution of the collected data is illustrated in Figure 5, where the regions with higher data concentration, particularly around hazard zones for the low-quality dataset, are highlighted in bright colors.

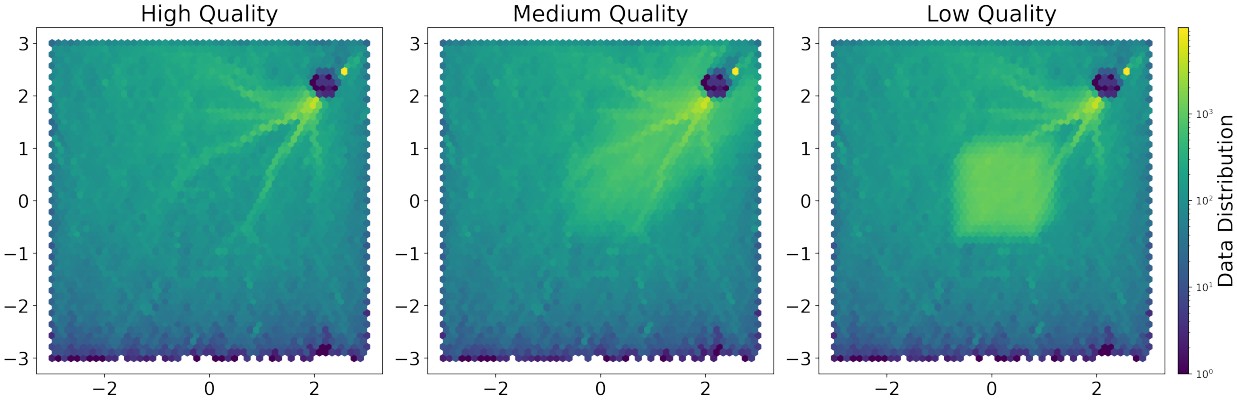

Figure 5: Data Distribution for High/Medium/Low Quality Offline Dataset

To more effectively demonstrate SEVPO's advantages across varying data quality scenarios (results in Table 1), we compared SEVPO with other baseline methods on Safety-Gymnasium tasks under three distinct data conditions: (1) a high-quality offline dataset predominantly composed of safe data, (2) a medium-quality offline dataset with a balanced mix of safe and unsafe data, and (3) a low-quality offline dataset largely consisting of unsafe data. To establish these scenarios, we created three different data distributions by selectively sampling the existing safe data and augmenting the unsafe data at the beginning of training. This approach resulted in safe-to-unsafe data ratios of 8:1, 1:1, and 1:3, respectively, with the exact data quantities shown in Table 6.

Table 6: Summary of Simulation Data Samples

| Task | Data Ratio | | Dataset | |
|---|---|---|---|---|
| | Safe | Unsafe | Number of Safe Samples | Number of Unsafe Samples |
| CarButton1 | 8 | 1 | 1,600,000 | 200,000 |
| | 1 | 1 | 1,600,000 | 1,600,000 |
| | 1 | 3 | 1,600,000 | 4,800,000 |
| CarButton2 | 8 | 1 | 1,600,000 | 200,000 |
| | 1 | 1 | 1,600,000 | 1,600,000 |
| | 1 | 3 | 1,600,000 | 4,800,000 |
| CarGoal1 | 8 | 1 | 560,000 | 70,000 |
| | 1 | 1 | 560,000 | 560,000 |
| | 1 | 3 | 560,000 | 1,680,000 |
| CarGoal2 | 8 | 1 | 1,600,000 | 200,000 |
| | 1 | 1 | 1,600,000 | 1,600,000 |
| | 1 | 3 | 1,600,000 | 4,800,000 |
| CarPush1 | 8 | 1 | 1,200,000 | 150,000 |
| | 1 | 1 | 1,200,000 | 1,200,000 |
| | 1 | 3 | 1,200,000 | 3,600,000 |
| CarPush2 | 8 | 1 | 1,600,000 | 200,000 |
| | 1 | 1 | 1,600,000 | 1,600,000 |
| | 1 | 3 | 1,600,000 | 4,800,000 |
| AntVelocity | 8 | 1 | 1,600,000 | 200,000 |
| | 1 | 1 | 1,600,000 | 1,600,000 |
| | 1 | 3 | 1,600,000 | 4,800,000 |
| HalfCheetah | 8 | 1 | 1,600,000 | 200,000 |
| | 1 | 1 | 1,600,000 | 1,600,000 |
| | 1 | 3 | 1,600,000 | 4,800,000 |
| SwimmerVel | 8 | 1 | 1,200,000 | 150,000 |
| | 1 | 1 | 1,200,000 | 1,200,000 |
| | 1 | 3 | 1,200,000 | 3,600,000 |

### C.2 Safety-Gymnasium

Safety-Gymnasium (Ray et al., 2019; Ji et al., 2023) is a highly modular and easily customizable benchmark environment library, built on MuJoCo, and designed to support research in Safe RL. The four distinct agents are "Car," "Ant," "HalfCheetah," and "Swimmer," while the tasks associated with these agents are "Button," "Goal," "Push," and "Vel." The numbers "1" and "2" indicate the difficulty levels of these tasks. Figure 6 provides visualizations of all the tasks.

### C.3 Evaluation Metrics

We employ normalized reward and cost as comparison metrics as proposed by (Liu et al., 2023a). Let $r_{\max}(M)$ and $r_{\min}(M)$ represent the maximum and minimum empirical reward returns for task M; detail values can be found in Table 7, respectively. The normalized reward is calculated as follows:

$$R_{\text{normalized}} = \frac{R_\pi - r_{\min}(M)}{r_{\max}(M) - r_{\min}(M)},$$

where $R_\pi$ indicates the evaluated reward return of policy $\pi$. In contrast, normalized cost is defined using a different formula to enhance result differentiation. It is computed based on the ratio between the evaluated

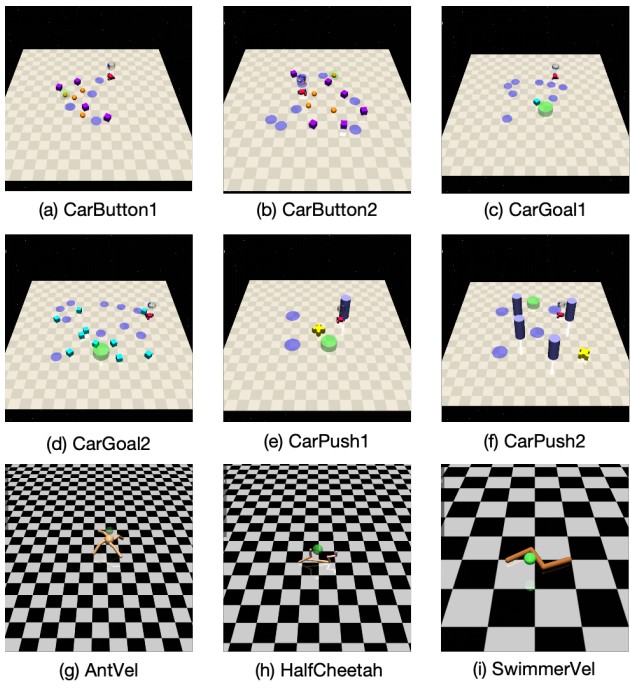

Figure 6: Safety-Gymnasium Visualization Overview

cost return $C_\pi$ and the cost limit $l$ :

$$C_{\text{normalized}} = \frac{C_\pi + \epsilon}{l + \epsilon},$$

where $\epsilon$ is a positive number to ensure numerical stability if the cost limit $l$ is 0. Note that the cost return and the cost limit are always non-negative. Therefore, both $C_{\text{normalized}}$ and $R_{\text{normalized}}$ fall within the range [0,1], where a lower cost is preferable and a higher reward is more desirable.

Table 7: Exact value for $r_{\max}$ and $r_{\min}$

| Task | $r_{\max}$ | $r_{\min}$ |
|---|---|---|
| CarButton1 | 44.42 | 0.0028 |
| CarButton2 | 41.99 | 0.0017 |
| CarGoal1 | 39.90 | 0.012 |
| CarGoal2 | 28.90 | 0.001 |
| CarPush1 | 16.30 | 0.013 |
| CarPush2 | 15.14 | 0.0002 |
| AntVel | 2976.27 | 6.15 |
| HalfCheetah | 2806.93 | 5.75 |
| SwimmerVel | 238.95 | 0.071 |

### C.4 Real-world Quadruped Navigation Application

To further evaluate the applicability of SEVPO, we conduct experiments using Nvidia Isaac Gym (Makoviychuk et al., 2021) with the Unitree Go2 quadruped robot. We design two dynamic safety-critical navigation tasks with varying difficulty levels: Simple and Hard configurations.

**Task Design**. Both tasks require the Go2 robot to navigate from an initial position to a designated destination point. In the Simple configuration, the environment features a flat terrain with stones placed at short intervals, requiring basic jumping maneuvers. The Hard configuration presents a more challenging scenario with steep slopes approaching 45 degrees, longer distances between obstacles, and higher failure

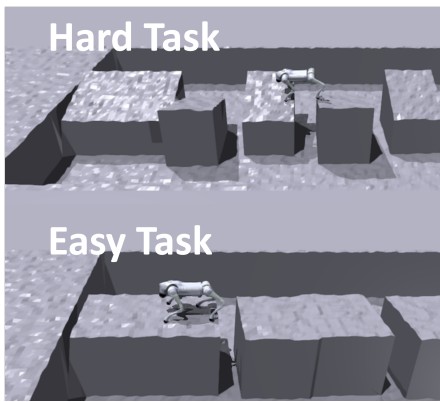

Figure 7: Navigation Task Overview

probability due to potential falls when the robot is inadequately trained. The overview of these two tasks is shown in Figure 7.

**Data Collection**. We employ the PPO algorithm to train an expert policy in an online setting, following (Cheng et al., 2024). The training process involves 8,192 parallel robot instances over 15,000 iterations. Using this well-performing policy, we collect expert demonstrations under both task environments. Additionally, we deploy a less-capable policy that can perform basic locomotion but fails to successfully jump across obstacles, generating low-quality trajectories that incur costs upon failure or missed jumps. The resulting offline dataset contains tuples of (states, actions, costs, rewards, next states, done).

**Dataset Composition**. Given the complexity of quadruped control compared to Safety-Gymnasium environments (Ray et al., 2019; Ji et al., 2023), substantial data is required for effective offline training. We collect 6,000 expert trajectories (average length $\approx$ 500 steps) and 3,000 low-quality trajectories (average length $\approx$100 steps, terminated early due to failures at obstacle crossings).

**Experimental Results**. In the offline training phase, we first evaluate both SEVPO and the baseline FISOR algorithm using only expert data (6,000 trajectories). Both methods demonstrate successful navigation and jumping behaviors. However, when trained on mixed data (6,000 expert + 3,000 low-quality trajectories), FISOR fails to maintain navigation performance and exhibits behavior mimicking the low-quality demonstrations, while SEVPO maintains robust performance despite the presence of suboptimal data. A demonstration video of both tasks can be found at `https://youtu.be/tDpWq2EV_Ig`.

### C.5 Hyper-Parameters

The entire training process for each task was conducted on a single RTX-4090 GPU, taking approximately two hours per task. All hyperparameters used in the experiments are detailed in Table 8.

**Training Details** We structured our training process into a two-stage approach to enhance efficiency. In the first stage, we train and save the critical components, $\check{Q}_c$ and $\check{V}_c$. In the second stage, we load these pre-trained components and proceed to update $Q_r$, $Q_c$, and $z_\theta$. This bifurcated training strategy significantly reduces the overall time required for model convergence. Throughout the training process, we employ the Adam optimizer and opt not to use weight decay regularization to avoid potential negative impacts on training dynamics. Different tasks require different parameters. For the loss weight controller $b_1, b_2$, in simpler task environments where learning a safe policy is easier, we prioritize maximizing rewards by setting $b_1 = 3, b_2 = 1$. Conversely, in more complex environments where learning a safe policy is challenging, we adjust the parameters to $b_1 = 1, b_2 = 3$ to emphasize safety. Additionally, we introduce an extra loss term as detailed in Eq. equation 15, and update the reference network $z'$ every $k = 5$ steps.

Table 8: Hyperparameters of BARS

| Parameter | Value |
|---|---|
| Activation function | ReLU |
| Expectile $\tau$ | 0.9 |
| Discount factor $\gamma$ | 0.99 |
| Soft update factor | 0.001 |
| $\alpha$ | 0.99 |
| Temperature $\mu_1$ | 3 |
| Temperature $\mu_2$ | 5 |
| Safe Loss Control $b_1$ | 1 |
| Unsafe Loss Control $b_2$ | 3 |
| Number of times Gaussian noise is added $T$ | 5 |
| Number of action candidates $N$ | 16 |
| Diffusion Model | DDPM (Hansen-Estruch et al., 2023) |
| $K_{\text{trust}}$ | 5e5 |
| Training steps | 1e6 |
| $k$ | 5 |

### C.6 Network Structure & Computation Cost

**Model components.** SEVPO maintains five neural networks in total: two conservative cost value networks, $(\check{V}_c, \check{Q}_c)$, which are used to identify and approximate safe-region boundaries, and three additional networks for policy optimization (see Section 4). The concrete network architecture is illustrated in Figure 8, and the corresponding update flow is illustrated in Figure 9.

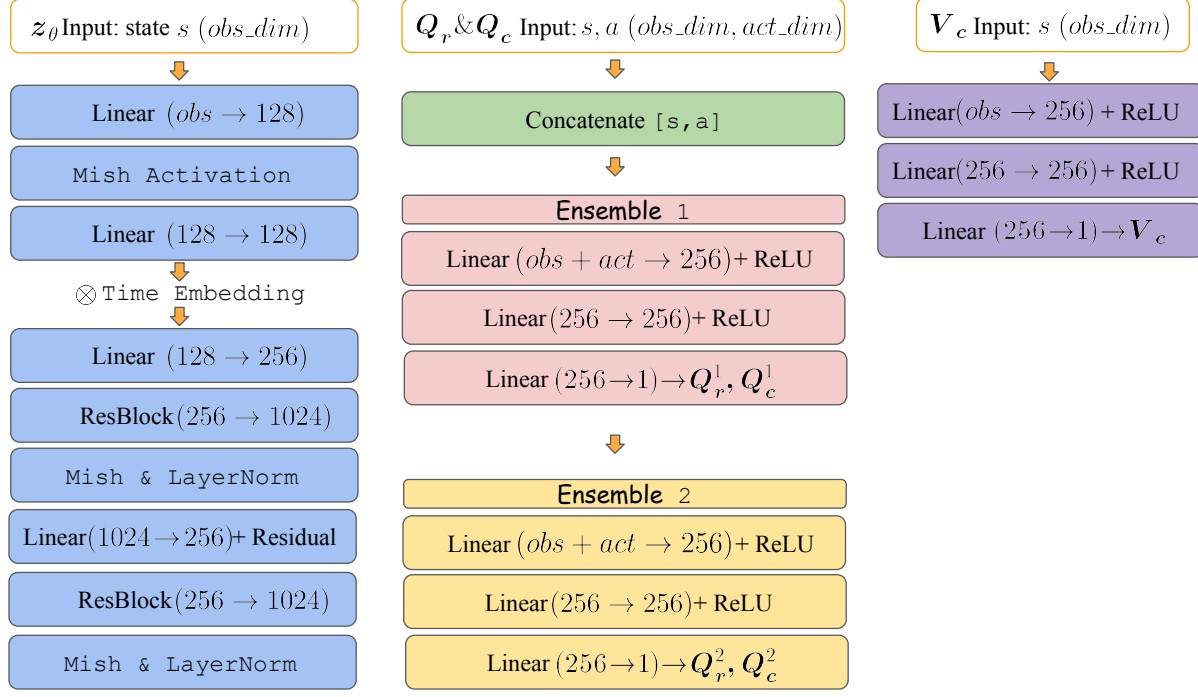

Figure 8: SEVPO Network Structure

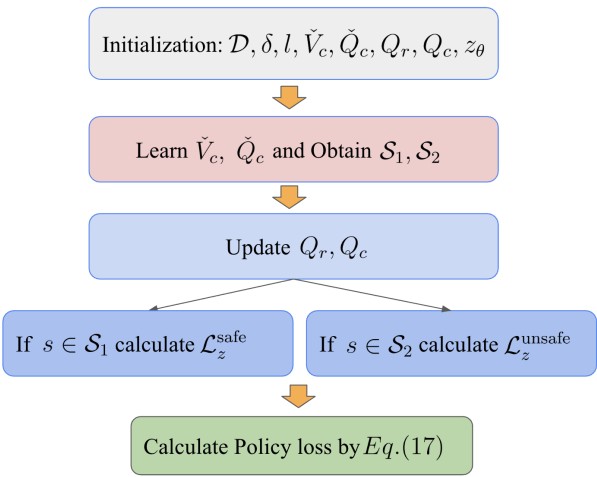

Figure 9: SEVPO System Structure

**Two-stage training protocol.** To reduce computational overhead, training is decoupled into two stages:

- **Stage 1 (Pretraining).** We train $(\check{V}_c, \check{Q}_c)$ only for cost-boundary estimation. This stage is executed once before policy optimization.

- **Stage 2 (Policy training).** We freeze $(\check{V}_c, \check{Q}_c)$ and update only the remaining three networks in the main training loop.

**Wall-clock time.** On CarButton1 with $10^6$ environment steps, SEVPO takes approximately **50 minutes** on a single NVIDIA RTX-4090 GPU, including **10 minutes** for Stage 1 and **40 minutes** for Stage 2. Under identical hardware and training settings, the baseline CPQ takes roughly **2 hours**, while FISOR takes roughly **45 minutes**. Overall, SEVPO achieves improved safety performance with only a minor wall-clock overhead compared to the fastest baseline, while remaining substantially faster than more computationally demanding alternatives.

# D   Ablation Studies

**Collect new data to elaborate policy gradually**  In this section, we outline the methodology for collecting new data using a pre-trained policy. Initially, we use the dataset from (Liu et al., 2023a) to randomly select a mini dataset for training our first agent, $\pi_1$, for $2 \times 10^5$ steps as a safe policy. Following this, we transition from an offline setting to an online environment for collecting new data. In this online mode, the agent interacts directly with the environment, executing the learned policy while observing rewards and any violations of predefined costs. These new data samples are then used to further train the policy for another $2 \times 10^5$ steps. We repeat this process for additional training rounds. In the experiment, we select a mini dataset for CarPush1, which includes 600,000 data samples, distributed across 600 trajectories, each containing 1,000 steps. Each data collection process is conducted using a high-performance RTX-4090 GPU, taking approximately 3 hours to gather a complete dataset for each task. The exact process is detailed in Figure 10.

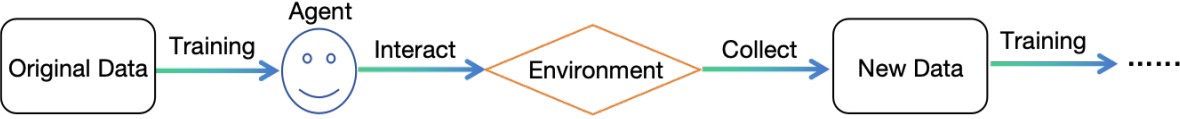

Figure 10: Training Process with New Dataset Collection

The newly collected data contains fewer instances of unsafe behavior, indicating that our trained agent is capable of executing safer policies. For the original CarPush1 dataset, the mean episode cost is 69.72. In comparison, the first collected dataset shows a significantly lower mean episode cost of 7.45, and the second dataset demonstrates further improvement with an even smaller cost of 4.41.

## E  Simulation Results and Learning Curves

The learning curves for the experiments are shown in Figure 11.

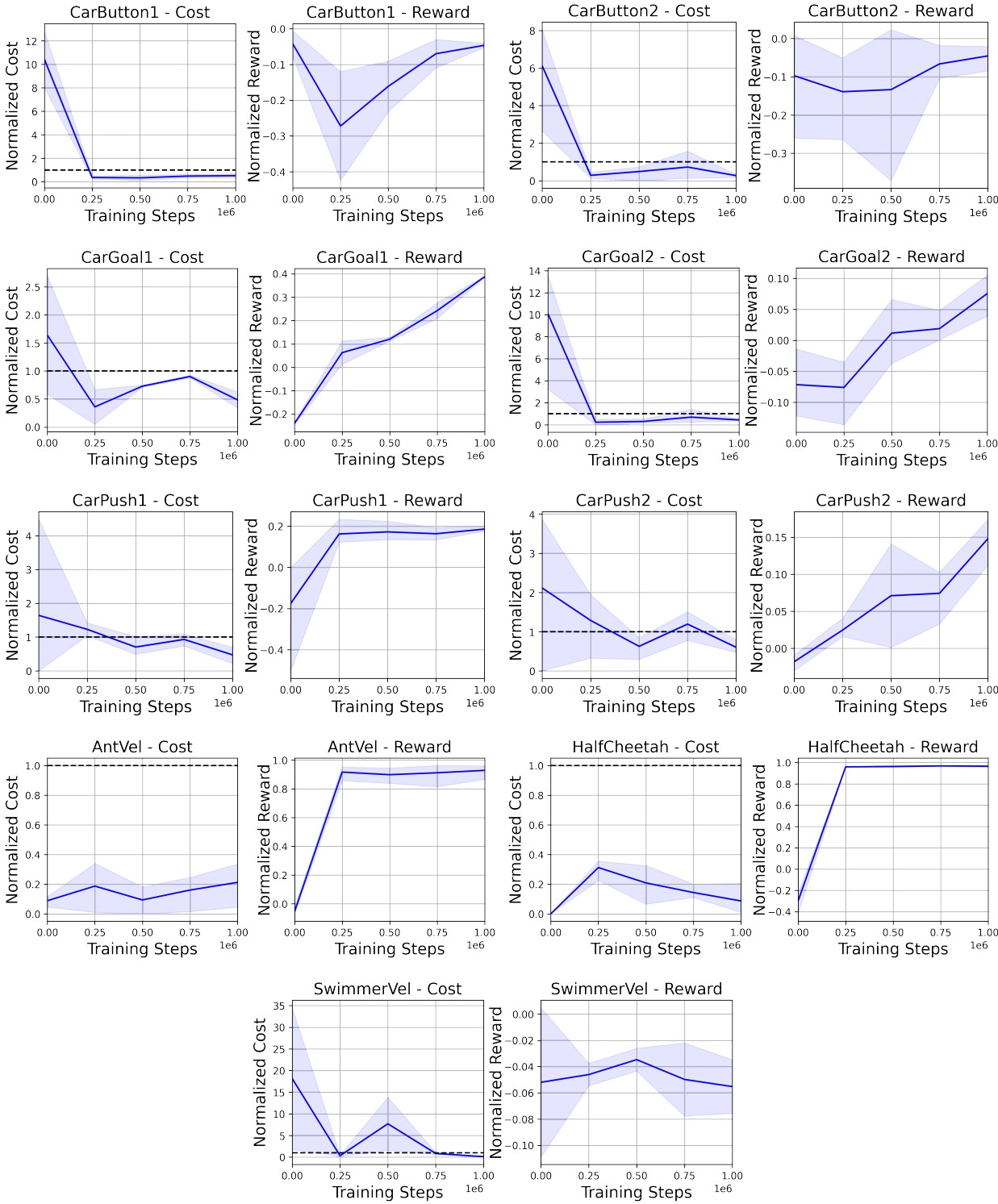

Figure 11: Learning Curves

