# OpenReview forum: "Divide and Conquer: Selective Value Learning and Policy Optimization for Offline Safe Reinforcement Learning"
_TMLR — Accepted by TMLR_

### Review · Reviewer_cd3H · 2026-01-07

**Summary Of Contributions:**

The submission proposes SEVPO, an algorithm for safe offline RL. The framework under which safe RL is defined is that of constrained MDPs, where the objective is to maximize cumulative rewards subject to a constraint on accumulated costs. The key insight the authors leverage is that the optimization can be split in two: in a region of the state space where it is infeasible to satisfy the cost constraint (even under the most conservative policy), it is desirable to find the policy that minimizes cost ("escape quickly"); in the complement of that region, the original constrained optimization problem persists, but is known to be feasible. The authors propose an algorithm that learns the safe/unsafe regions by applying implicit Q-learning (IQL) on the problem of minimizing costs: if the state value of the most conservative policy is above the threshold, then a state is "unsafe", else it is "safe". Within the safe region, the authors follow the FISOR approach of Zheng et al. (2024), which learns a diffusion policy that maximizes a trade-off between rewards and costs, subject to a behavior regularization constraint that restricts the policy to lie close to the behavior policy that generated the data. In the unsafe region, the diffusion policy is optimized to minimize cost.

######## Strengths ########

- The main strength of this submission is the performance of the proposed algorithm. Across the Safety-Gymnasium tasks from DSRL, the proposed algorithm is safe (on average) in all tasks, and achieves the highest rewards among safe agents in all but 2. The algorithm also performs well on a realistic robot locomotion simulation on Isaac Gym compared to an unidentified baseline.
- The writing of the manuscript is for the most part clear and easy to follow. The example in Figure 2 is excellent for explaining to the reader the challenges of learning under the two constraints of behavior regularization and safety.
- My sense that the proposed approach is sensible for solving the safe offline RL problem and that the improvements of SEVPO over baselines are justified by the choices made in the algorithm's design. (Note that my confidence in this is not very high, for the reasons outlined in the remainder of this review.)

######## Weaknesses ########

- The main weakness is the lack of clarity on the similarities and connections of the proposed SEVPO approach and FISOR (Zheng et al., 2024). In particular, these similarities are only discussed in Appendix A.1, which incidentally is never referenced in the main paper.
- It is often difficult to identify in the authors' descriptions which improvements incorporated in their approach are equally useful for the safe online RL setting vs which are specific to the offline setting. In particular, in Section 2:
    - "Many safe RL algorithms consider [Eq. 5]... However, this formulation faces several issues in the offline safe RL setting: (i) ... Eq. 5 enforces the safety budget only in expectation" -- Is this any different in the online case? Or does this limitation apply equally in both online and offline safe RL? It is worth clarifying these distinctions and ensuring that the reader is made aware of which challenges affect safety in general vs in the specific case of offline RL.
    - I also have this doubt about (ii), but I believe that in the online case the infeasibility can be overcome with additional rollouts because $\pi_\beta$ changes over time. Is that what makes (ii) relevant to offline (but not online) safe RL?
- The reasoning behind many design choices in the algorithm is either omitted or insufficient.
    - Section 3: Per remark 3, The key piece of insight is that excluding policy regularization outside the safe region removes the possibility of making the problem infeasible.
        - But I can't quite follow the logic
        - In the unsafe region, there are no policies that satisfy the safety constraint, and so clearly we need to drop the safety constraint in that region. But once that constraint is dropped, there is no potential for infeasible solutions: one could minimize $V_c^\pi$ subject to the behavior regularization constraint. An argument made earlier in the paper is that we don't want to constraint the agent to behaviors in the data: we simply want to get out of the unsafe region as quickly as possible. But this is not related to feasibility
        - In the safe region, by definition there exist policies that satisfy the safety constraints. But how does this guarantee that at least one such policy also satisfies the behavior regularization constraint? I believe this is not the case.
        - My take is that separating the two problems reduces the potential for infeasible solutions, but it is still possible for infeasible solutions to arise inside the safe region.
        - Overall, my suggestion would be to comment on these nuances throughout the presentation, and to avoid making such strong claims about what the separation into safe/unsafe regions grants us.
        - It also seems to me that, just like IQL is used to remove the need for behavior regularization in the unsafe region, it would be possible to use IQL to remove the need for behavior regularization in the safe region. Instead, the authors use standard Bellman updates in the safe region and incorporate behavior regularization.
            - One alternative reasoning is that the behavior regularization is kept in for safety, which is common in online safe RL in order to prevent the agent from stepping too far when creating new roll-outs to encourage safe exploration. But here safety is already accounted for in the data (we've estimated that this is in the safe region, and we could constrain the search to that safe region via the safety constraint and the OOD training via IQL).
    - Section 4.2
        - The optimization that results from Eq. 11 is no longer equivalent to the CMDP optimization problem. It goes beyond maximizing reward s.t. a cost constraint and instead maximizes a trade-off between rewards and costs s.t. the same cost constraint. The authors arguments do not address this fundamental disconnect, but instead provide a heuristic justification (discourage unsafe actions, mitigate critic estimation errors, "guarantee" zero violations.)
        - It is unclear to me why Eq. 13 uses $\check{Q}$ and $\check{V}$, which are not the values for the learned policy. The goal should be to minimize the cumulative sum of costs under the current policy, not the conservative policy (which will not be followed after the agent exits the unsafe region)
        - The optimization in Eq. 15 is not clear. How do the authors optimize the diffusion policy where the loss only contains the action sampled after solving the ODEs/SDEs? Since this is the main conceptual difference between the proposed SEVPO approach and FISOR (Zheng et al., 2024), it would be critical to more explicitly explain this point.

**Audience:**

Yes

**Audience Explanation:**

The manuscript addresses an important and timely topic, contains algorithmic novelty, and the approach performs better than state-of-the-art approaches (especially in terms of ensuring safety).

**Claims And Evidence:**

No

**Claims Explanation:**

Many of the claims are well-supported, but I have two main concerns.

- The description throughout sections 1 through 4 suggest that the structure of SEVPO, and in particular the separation of the state space into a "largest safe" and an "smallest unsafe" regions, are novel. However, this separation, the learning of these regions from offline data, the objective functions for the two regions, the use of diffusion policies and the formulation of the diffusion objectives all follow the structure of FISOR (Zheng et al., 2024).
    - Appendix A.1 is the first time in the manuscript that the authors fairly characterize the similarities between FISOR and the proposed SEVPO approach.
    - The distinctions are in the details: learn the separate regions via IQL instead of HJ reachability and remove the behavior regularization within the unsafe region. While these are fine innovations and indeed make SEVPO distinct from FISOR, the manuscript should explicitly claim, up front, that the structure of the algorithm mimics that of FISOR and that the proposed approach is a modification/adaptation/extension/improvement of FISOR.
    - One more note on Appendix A.1: As far as I understand, the only theoretical result that arises from dropping the behavior regularization from the unsafe region is that the resulting policy achieves minimum cost within the unsafe region. This seems largely trivial given that this is the problem optimized within the unsafe region.
        - Also note that this is the *smallest* unsafe region
- The results displayed on Table 1 are very strong, and the paper's strongest argument. However, the authors' characterization of these results throughout the paper (starting from the abstract) appears incorrect. In particular, the manuscript claims that SEVPO performs well in settings where the dataset contains varying ratios of safe-to-unsafe data. However, as revealed in Section 5, this ratio is controlled by "duplicating unsafe [trajectories]". Duplicating unsafe trajectories is not equivalent to changing the safe-to-unsafe ratio in the data, but rather upweighting the importance of unsafe vs safe data in the objective function---this is because the additional unsafe data observed by the algorithm consists of duplicate data points. Upsampling is a standard technique for optimizing weighted objectives. This makes it so that all rows beyond "ori" (original ratio) in Table 1 are really measuring the various algorithms' abilities to learn good policies as the weighting of the safe vs unsafe trajectories varies. It is unclear to me how this may be confounded with the $b_1$ and $b_2$ weights in Eq. (17).

**Requested Changes:**

The key changes required to make the paper suitable for publication are:
- Clarify the strong connection of SEVPO and FISOR throughout the manuscript (and not just in Appendix A.1) and being up-front about what are the key innovations of SEVPO as compared to FISOR.
- Ensure that the distinction between changes that are useful for safe RL in general vs for safe RL in the specific context of offline RL is clear.
- Provide adequate justification for the design choices behind SEVPO discussed in the "Summary" box under "Weaknesses"
- Adapt the discussion of the results in Table 1 to remove the claim that the results represent various safe-to-unsafe data rations, or adequately justify why their experimental setup does demonstrate what they claim.

######## Additional feedback ########
The following points are provided as feedback to hopefully help better shape the submitted manuscript, but will not impact my recommendation in a major way.

Abstract
- "Existing methods extend offline RL with primal–dual value learning and behavior-regularized policy optimization" -- to what end? Authors justify why this doesn't address safety, but do not mention why someone would want to "extend offline RL" this way in the first place
- Must clarify that the Go2 experiments are in simulation

Intro
- "However, safe offline RL introduces challenges that go well beyond those in standard offline RL" -- this is a weak argument for why existing approaches that explicitly constrain the policy are insufficient for the safe offline RL setting. What specifically about those prior methods is insufficient?
- "In practice, this means that policies may satisfy constraints in expectation while still violating safety at specific states" -- this is a strong and well-worded argument. However, I do not believe that the experiments support that SEVPO addresses this.
- Paragraphs starting with "Mitigating the impact of OOD actions" and "Most existing methods" seem to belong in a separate "Related Work" section. Consider including only a brief summary in the introduction as a motivation for the approach, and then the current detailed version in a stand-alone section.
- "Figure 1 illustrates this..." -- Fig 1 doesn't really illustrate how "SEVPO selectively applies policy regularization..." Instead, it illustrates the results of applying the approach.
- Item one in the contributions list suggests that the method assumes that unsafe regions are characterized by high costs. While this may seem natural to folks familiar with the constrained MDP formulation from Section 2, it may be worth explicitly clarifying that this is a standard assumption which will be detailed in Section 2.
- The train-collect-train framework is a nice contribution, especially in the context of safety aware RL. I wish it had been mentioned much sooner.

Sec 2
- "the optimal policy depends on the initial state" -- I didn't know this, and it seems non-obvious. This becomes significant because the authors then focus on a fixed initial state. The question is: what would need to change (in the analysis, in the derivations, in the approach) to handle the case with varying initial state?
- Paragraph "Out of Distribution in Offline RL" is almost just a repetition of the final few sentences of paragraph "Offline reinforcement learning." Consider consolidating into one.
- "Formally, [the loss function minimized by] these methods..."
- Fig. 2 example
    - Great example, but maybe the writing could benefit from a few clarifications.
    - "... reaches $s_5$ while minimizing cost violations" --> "while limiting cost violations to a budget $l$".
    - Presumably the discount factor here is $\gamma=1$?
    - I did not get from this example what a "bad" state would be, where violations are unavoidable. Including such a state in this example would be helpful.

Sec 3
- "In the safe region, the objective is to maximize..." -- Consider explicitly stating that this is the standard safe online RL objective from Eq. 5.
- "any trajectory starting in these states will inevitably violate the safety budget" -- "inevitably" seems incorrect, as it depends on the budget and how far from the safe region the agent is (in terms of # actions). Consider "potentially"
- It's fine to relegate the proof of theorem 4 to the appendix, but I would encourage the authors to include a one/two-sentence summary of the arguments of the proof (e.g., "$\pi_2$ dominates $\pi_1$ because $\pi_1$ is a feasible solution to the optimization of Eq. 6, and Eq. 5 does not have a feasible solution in the unsafe set and so the min cost solution is $\pi_2$.")

Sec 4
- Did the authors mean $\check{Q}_c$ and $\check{V}_c$ instead of $\hat{Q}_c$ and $\hat{Q}_v$?

Sec 4.1
- It's possible that this is stated and I just missed it, but it is worth explicitly stating that the values learned in Eq. 7/8 are the ones obtained for the policy that minimizes cost across all states. I believe that this is what the authors mean to convey with the "most conservative policy", but for clarity, it may be worth spelling it out.
- It may also be worth explicitly stating that this is distinct from the minimization problem for $Q_\pi^C$ in that $Q_\pi^C$ applies only to states in the unsafe region, where $\pi$ is different from the most conservative policy in states in the largest safe region.

Sec 4.2
- It is unclear to me what the authors' purpose is in introducing the "valid distribution" constraint in Eq. 13. Isn't this a trivial constraint that is satisfied by choosing an appropriate policy parameterization (e.g., Gaussian, diffusion, categorical...)?
- Diffusion parameterization
    - Looking at Zheng et al. (2024), I believe that the indicator function for $w_\text{safe}$ should be outside of the exponential (otherwise, you're saying: imitate the behavior policy exactly when Q_c exceeds threshold, since (exp(0)=1).
    - I seems that the argument for why the weighted objective in Eq. 14 makes sense is in Zheng et al. (2024), but the submitted manuscript should be self-contained and explain that this corresponds to the optimal solution, as derived in Zheng et al.

Sec 5
- Results are actually very strong, even if we only consider the "original" safe-to-unsafe ratio
- It is worth noting that Table 2 shows that the policy restriction increases the cost, while maintaining it below the threshold of 1. [Also, please make sure that the table caption is  more descriptive (e.g., "Normalized results for the ablation studying the impact of policy restriction within the unsafe region")]
- "Real world" is not a correct description of Isaac Gym.

Typos/style/grammar
- "Isaac Gym(Makoviychuk et al., 2021)"
- "actor-critic" and "primal-dual" are written with an en-dash ("--") instead of a hyphen. Is this deliberate? (I've seen this happen often when copy-pasting content generated by LLMs, which sometimes directly uses the en-dash symbol.)
- Sec 2, point (i): "Eq. equation 5"
- The placement of Fig 2 is quite bad, as it is inline next to an unrelated paragraph in a different section.
- Sec 3: "Eq.(3)" -> "Eq.~(3)" (there are many instances of this)
- Page 7, before Eq. 9: "bellman" -> "Bellman"
- Sec 5: "Eq.equation 16"

---

### Review · Reviewer_5RLU · 2026-01-16

**Summary Of Contributions:**

The paper accurately identifies the core challenges in offline safe reinforcement learning: existing methods' uniform treatment of all states leads to inaccurate value estimation, infeasible solutions when constraints conflict, and sensitivity to data quality. To solve this, the paper presents the divide-and-conquer approach, SEVPO, which separates value learning and policy optimization for safe and unsafe regions.

**Audience:**

Yes

**Audience Explanation:**

The divide-and-conquer approach makes some sense.

**Broader Impact Concerns:**

None.

**Claims And Evidence:**

Yes

**Claims Explanation:**

The paper proves through Theorem 4 that the optimal solution of the decoupled optimization problem is superior to that of the traditional formula, providing theoretical support for the method. At the same time, a simple MDP example (Figure 2) intuitively illustrates the failure mode of the traditional method.

**Requested Changes:**

1. There is a lack of a clear network architecture diagram and a computation cost.
2. About Table 5, Lack of sensitivity analysis for multiple hyperparameters.
3. About the discussion about the safe region, how to define a safe region in a new environment when transferring this method.

---

### Review · Reviewer_9Wbg · 2026-02-01

**Summary Of Contributions:**

Summary\
This paper addresses challenges in offline safe reinforcement learning, specifically constraint infeasibility and performance degradation when safe and unsafe data are mixed. The goal is to learn policies that balance rewards and cost constraints using only offline data. The authors propose SEVPO, which partitions states by safety. It first uses a conservative policy to learn cost values, delineating the largest safe region. Subsequently, within the safe region, reward optimization proceeds with behavioral constraints. In unsafe regions, the focus shifts to cost minimization, encouraging a return to the safe zone. Experiments in Safety Gymnasium and quadruped navigation simulations demonstrate SEVPO's more robust cost reduction and higher reward maintenance compared to baselines. The method's motivation is clear, and its design reasonable. Evidence is generally sufficient but could be strengthened regarding safety evaluation and real-world task quantification. Overall, this partitioned offline safe optimization method shows novelty and engineering value. Further refinement of metrics and discussion of related work are recommended to enhance persuasiveness.\

Strength\
1.The paper focuses on a core challenge in offline safe reinforcement learning: the coexistence of safe and unsafe experiences in the dataset. This makes the research motivation very clear and practically significant.\
2.The proposed partitioned objective design separates value learning and policy optimization into safe and unsafe regions. This design is intuitive and aligns with the actual needs of different states, effectively avoiding conflicts and training instability that a single objective might cause.\
3.Introducing behavioral regularization in the safe region aids stable policy learning. Removing it in the unsafe region encourages the system to explore optimal, low-cost escape paths. This targeted policy design is logically clear and rational.\
4.The experimental setup is comprehensive, systematically comparing various mainstream methods on Safety Gymnasium tasks from the DSRL benchmark. By constructing scenarios with different ratios of safe and unsafe data, the robustness of the proposed method is thoroughly verified.\
5.Using a Unitree Go2 quadruped robot for a navigation task in the NVIDIA Isaac Gym simulation environment, the research demonstrates its engineering practicality and potential application value.\

Weakness\
1.The paper primarily presents safety evaluations through normalized costs and expected rewards. However, in safety-critical applications, direct quantitative descriptions for worst-case risks, failure frequency, or extreme events (e.g., maximum duration exceeding safety thresholds) are lacking. This may reduce the persuasiveness of safety claims.\
2.Identifying safe regions relies on cost value estimation and predefined thresholds. Insufficient sensitivity analysis exists regarding these critical parameters. For example, how different threshold settings or estimation errors affect safe region boundaries and their robustness requires further argumentation to strengthen the method's credibility.\
3.The description of the quadruped robot navigation experiment's task and data construction is relatively detailed. However, the quantitative presentation of experimental results could be richer. Supplementing success rates, fall rates, detailed breakdowns of different path costs, and performance variance statistics from multiple runs would help reviewers more rigorously assess its practical effectiveness and advantages.\
4.The related work section could be enhanced by citing and discussing recent research on data quality robustness, generalization ability of models on unseen data, and algorithms for adapting to changes in data distribution. \
Suggested references include:\
Graph Anomaly Detection with Few Labels: A Data-Centric Approach (KDD 2024)\
Type-LDD: A Type-Driven Lite Concept Drift Detector for Data Streams (IEEE TKDE 2024)\
Boosting the Adversarial Robustness of Graph Neural Networks: An OOD Perspective (ICLR 2023)\

Question\
1.How can this framework adaptively adjust the division between safe and unsafe regions to effectively cope with environmental changes?\
2.SEVPO performs well with varying data quality. Could a mechanism be explored to enable it to dynamically assess data quality during training and optimize the partitioning strategy accordingly?

**Audience:**

Yes

**Audience Explanation:**

Offline reinforcement learning and safety-constrained learning are current hotspots in machine learning. They hold significant research and application value, especially in scenarios involving high risk and cost interactions. This paper's proposed state-partitioned offline safe optimization strategy innovatively addresses challenges faced by traditional methods in mixed data environments. Through validation in benchmark tasks and robot simulations, the method demonstrates robustness under data uncertainty and declining quality. These characteristics align well with the TMLR audience's interest in methodological novelty, theoretical soundness, experimental reproducibility, and practical application potential. Therefore, this paper's findings will be highly attractive to TMLR readers engaged in safe reinforcement learning, offline learning, robot control, and algorithms focused on data quality robustness.

**Broader Impact Concerns:**

The potential positive impact of this paper is that, through the SEVPO framework, policies with more reliable safety performance can be trained, even with mixed and potentially low-quality offline data. This is crucial for high-risk, high-cost real-world systems such as autonomous driving, industrial robots, and medical assistance systems. It could significantly reduce personal injury and equipment damage risk during training, thereby accelerating safe reinforcement learning technology application in practice.\
Regarding potential risks, offline data may suffer from incomplete coverage (especially in extreme or uncommon states), imperfectly defined cost functions, or subtle environmental changes after deployment. In these cases, even SEVPO-trained policies may exhibit unpredictable "tail risks" in insufficiently covered risky areas. Therefore, in the broader impact section, it is recommended to add explanations on how to identify data biases, conduct rigorous pre-deployment verification processes, analyze potential failure modes, and emphasize human monitoring and intervention mechanisms. This will help prevent misuse and reduce risks from direct deployment without adequate safety assurances.

**Claims And Evidence:**

Yes

**Claims Explanation:**

The paper's core argument is that the state partitioning method effectively solves infeasibility and instability issues caused by a single objective in offline safe optimization. It maintains stronger safety and performance even when data quality declines. The proposed method and research motivation are highly aligned. Through partitioned objectives and selective regularization design, the paper directly addresses the distinct needs of safe and unsafe states.\
In terms of experiments, the paper systematically compares various advanced methods in Safety Gymnasium tasks. It innovatively constructs three data distribution scenarios: high, medium, and low quality. Experimental results clearly show that as the ratio of safe to unsafe data changes, the proposed method more effectively keeps normalized costs below the safety threshold and achieves competitive rewards. Furthermore, the quadruped robot navigation simulation provides preliminary verification of the method's potential application in complex real-world scenarios.\
It should be noted that current safety proofs primarily rely on expected values. Supplementing direct assessments of worst-case risks and failure frequencies, such as maximum single-step cost, would more intuitively demonstrate risk levels and further enhance the reliability of safety claims.

**Requested Changes:**

1.It is recommended to supplement more specific safety evaluation metrics, such as maximum single-step cost or the number of consecutive constraint violations, to more comprehensively demonstrate the policy's safety performance in extreme situations.\
2.It is suggested to conduct a sensitivity analysis on key parameters used in safe region identification to understand how different parameter settings affect safe region delineation and their impact on overall performance.\
3.It is recommended to expand the discussion in the related work section to include recent research on improving data quality robustness, enhancing models' generalization capabilities on unseen data, and mechanisms for adapting to changes in data distribution. This would more clearly articulate the paper's innovations and distinctions from existing work.

---

### Decision · Action_Editor_KR6U · 2026-04-01

**Recommendation:** Accept as is

**Audience:**

Yes

**Audience Explanation:**

readers from safe RL and offline RL fields would be interested in this

**Claims And Evidence:**

Yes

**Claims Explanation:**

reviewers agree that evidence is accurate, convincing, and clear